# DNA targeting by compact Cas9d and its resurrected ancestor

Rodrigo Fregoso Ocampo[1], Jack P. K. Bravo [2,6], Tyler L. Dangerfield [2], Isabel Nocedal[3], Samatar A. Jirde[3], Lisa M. Alexander [3], Nicole C. Thomas[3], Anjali Das[2], Sarah Nielson[1], Kenneth A. Johnson[1,2], Christopher T. Brown [3], Cristina N. Butterfield[3], Daniela S. A. Goltsman[3] & David W. Taylor [1,2,4,5] ✉

Type II CRISPR endonucleases are widely used programmable genome editing tools. Recently, CRISPR-Cas systems with highly compact nucleases have been discovered, including Cas9d (a type II-D nuclease). Here, we report the cryo-EM structures of a Cas9d nuclease (747 amino acids in length) in multiple functional states, revealing a stepwise process of DNA targeting involving a conformational switch in a REC2 domain insertion. Our structures provide insights into the intricately folded guide RNA which acts as a structural scaffold to anchor small, flexible protein domains for DNA recognition. The sgRNA can be truncated by up to ~25% yet still retain activity in vivo. Using ancestral sequence reconstruction, we generated compact nucleases capable of efficient genome editing in mammalian cells. Collectively, our results provide mechanistic insights into the evolution and DNA targeting of diverse type II CRISPR-Cas systems, providing a blueprint for future re-engineering of minimal RNA-guided DNA endonucleases.

To cleave a specific site in dsDNA, Type II CRISPR-Cas nucleases associate with a single guide RNA (sgRNA), a chimeric CRISPR RNA (crRNA)-trans-activating crRNA (tracrRNA), targeting a region complementary to the first 18-24 nucleotides of the 5′ end of the sgRNA, termed the spacer[1–6]. These nucleases cleave the target and non-target strands using the HNH and RuvC endonuclease domains, respectively[7]. It is widely thought that type II endonucleases evolved from IscB systems, originating from IS200/IS605 transposons, which contain a small endonuclease (420–500 amino acids in length) and a large obligate mobile element guided activity RNA (ωRNA)[8]. As IscB was co-opted in bacterial adaptive immune systems[9], the protein domains have increased in size while their guide RNA (sgRNA) has decreased[10]. While the closest homolog to IscB that has been structurally characterized to date is the type II-C endonuclease, *Campylobacter jejuni* Cas9 (Cj.Cas9) (987 amino acids and 97-nt long sgRNA), structural insights into other evolutionary intermediates

between IscB and type II-C CRISPR nucleases have remained elusive[11,12].

Recently, type II-D CRISPR-Cas systems were discovered through metagenomic analysis of uncultivated microbes[3]. These systems are encoded in genomes as a CRISPR array adjacent to a gene coding for a Cas9 endonuclease. Type II-D CRISPR systems form a distinct phylogenetic cluster[12,13] separated from other systems, and lack the Cas1, Cas2 and Cas4 adaptation genes[14–17]. Furthermore, type II-D Cas9 nucleases are also smaller than other type II nucleases ranging from ~600-970 amino acids, compared to 1200-1400 for type II-A systems but with larger sgRNA requirements (crRNA:tracrRNA)[18,19]. A previously characterized 747 amino acid Cas9d from a clade identified as MG34 (Cas9d-MG34-1) requires a 158-nt sgRNA and is an active nuclease in vitro and in *E. coli*. Altogether, the small RNP size, the shorter spacer requirement (18-20-nt as opposed to 20-24-nt) and the ability to produce staggered DNA ends after cleavage, highlights the diverse and

[1]Interdisciplinary Life Sciences Graduate Programs, University of Texas at Austin, Austin, TX 78712, USA. [2]Department of Molecular Biosciences, University of Texas at Austin, Austin, TX 78712, USA. [3]Metagenomi, Inc., 5959 Horton St, Floor 7, Emeryville, CA 94608, USA. [4]Center for Systems and Synthetic Biology, University of Texas at Austin, Austin, TX 78712, USA. [5]LIVESTRONG Cancer Institutes, Dell Medical School, University of Texas at Austin, Austin, TX 78712, USA. [6]Present address: Institute for Science and Technology Austria (ISTA), Klosterneuburg, Austria. ✉e-mail: dtaylor@utexas.edu

unique properties of these nucleases[20–22]. While the related Cas9d-MG102 family of nucleases has shown potential for human genome editing applications due to their activity in mammalian cells[3], a significant challenge for therapeutic development has been a lack of activity in mammalian cells for the smallest Cas9d nucleases. The Cas9d-MG34 family is quite rare compared to other Type II nucleases and no activity has been observed in mammalian cells for the few members of this family to date.

Here, we performed biochemical and structural characterization of Cas9d-MG34-1 (hereby referred to as Cas9d) ribonucleoprotein (RNP) to understand the architecture and mechanisms governing DNA targeting activity for these systems. We obtained a series of structures that reveal mechanistic insights into sgRNA and target DNA recognition by Cas9d. Structural comparison of Cas9d with other Cas9 nucleases and the ancestral IscB reveals a common ribonucleoprotein core furnished with increasingly large protein regulatory domains, which were acquired during evolutionary specialization[23,24]. Our findings demonstrate how structure-guided sgRNA truncation can be used for engineering these systems to generate streamlined gene-editing technologies. We use ancestral sequence reconstruction to generate a synthetic sequence representing a putative ancestral Cas9d nuclease that is about 200 amino acids smaller than members of the MG102 family and exhibits endonuclease activity in mammalian cells, with indel formation greater than 50% in certain targets. These results demonstrate the potential of programmable nucleases from this family to expand in vivo human genome editing applications, including by optimizing delivery through cargo-size-limited technologies.

## Results

### Structure of the Cas9d-sgRNA-target DNA complex

We first co-expressed the Cas9d with its sgRNA (Supplementary Fig. 1a, b), reasoning that this would increase stable complex assembly. We biochemically validated sgRNP assembly using in vitro plasmid targeting assays without additional sgRNA supplementation (Supplementary Fig. 1c). Cas9d efficiently cleaved its target plasmid, confirming accurate assembly of the enzyme with its sgRNA. We then determined the cryo-EM structure of a Cas9d–sgRNA–target dsDNA complex in its product state at a nominal resolution of ~2.7 Å (Fig. 1a–c, Supplementary Fig. 2a, b and Supplementary Fig. 3a–d).

Our structure revealed that Cas9d adopts a bilobed architecture like other type II endonucleases[25,26]. However, neither of the endonuclease domains were resolved in this structure, suggesting that they are both disordered after dsDNA cleavage or more generally sample multiple conformational states. While the conformational heterogeneity of the HNH domain in other Cas9 endonucleases has been well-documented, intriguingly the RuvC domain also showed flexibility in the post-cleavage state.

Similarly to its closest evolutionary homologs, type II-C nucleases, including Cj.Cas9 and Cd.Cas9, both REC2 and REC 3 domains form multiple contacts with the PAM proximal and PAM distal R-loop, respectively, while the REC1 domain is wedged between the sgRNA repeat anti-repeat stem loop[27,28] (Supplementary Fig. 4a, b). Consistent with other Cas9 endonucleases, Cas9d has an Arginine-rich bridge helix which helps stabilize the PAM proximal region of the sgRNA (Supplementary Fig. 4c). The PAM interacting region has a highly electropositive surface, which stabilizes interactions with the phosphate backbone of the dsDNA, as well as the first base pairs of the repeat anti-repeat P1 region of the sgRNA[29]. While our Cas9d structure exhibits hallmarks of other Cas9 endonucleases, its unique sgRNA fold and small flexibly tethered domains provide a structural rationale for the distinct phylogenetic clustering of type II-D endonucleases.

### Cas9d sgRNA provides a scaffold for protein domains

Cas9 nucleases consist of several globular domains tethered by flexible linkers. This feature facilitates drastic conformational changes upon sgRNP complex formation and R-loop progression, which is critical for the control of Cas9 nuclease activity[26,30]. Similarly, these globular domains tethered by flexible linkers are intertwined with the folded sgRNA (Fig. 1c–e). The REC domains are enriched in basic residues that make numerous contacts with the sgRNA backbone. While these contacts lack sequence specificity, they appear to be highly specific for RNA tertiary structure, enabling the sgRNA to effectively pre-organize the protein domains for DNA targeting.

The sgRNA is intricately folded, comprising four distinct paired regions (P1-P4). P1 contains the repeat-anti-repeat (or nexus) stem-loop, a 17-bp helix with a GAAA tetraloop (Fig. 2a). Like other type II systems, this region contacts both the REC1 and PI domains to stabilize the sgRNA–dsDNA complex[31]. The sgRNA P2 and P3 helices fold over each other via base stacking and H-bond interactions (Supplementary Fig. 5a–c) and occupy the region that REC2 and REC3 occupy in other type II systems. A78 and A80 base stack in the interface of the P2 and P3 helices and non-canonical H-bonding interactions between the exocyclic amine in G77 and O4 in G125 (Fig. 2b, c) further stabilize the intricate P2-P3 fold. Additionally, the sgRNA forms a triplex structure comprised of A82, U110, and C107 (Fig. 2d). The P4 region is comprised of a 4-bp region connected by a tetraloop linking the P2 and P3 regions. P3 and P4 separate P2 into two segments (nts 60-79 and 125-139). In contrast with secondary structure predictions, nucleotides 142-146 are not unstructured and instead form a long-range kissing loop interaction with P3.

### Cas9d sgRNA can be trimmed to streamline the nuclease

To test the importance of different regions of the sgRNA we used an in vivo DNA targeting assay in which Cas9d is co-expressed with its sgRNA in a two-plasmid system, enabling us to mutate the sgRNA independently of the endonuclease, while assessing activity in E. coli. We first trimmed the entire P4 region (ΔP4) and the last 7 bps of the P1 RAR region (ΔP1), which did not form any contacts with the endonuclease (Fig. 3a). As expected, these mutations did not affect DNA targeting in vivo when trimmed individually (Fig. 3b). Next, we performed a set of combinatorial mutations by trimming the unresolved nucleotides in our structure (ΔP3) and combining them with the previously performed mutations ΔP1ΔP3, ΔP4ΔP1, and ΔP1ΔP3ΔP4. Both double mutants retained targeting activity. Notably, the triple mutant (ΔP1ΔP3ΔP4) reduced the size of the sgRNA by ~25% without compromising DNA targeting activity (Fig. 3b). This minimal RNP is smaller than OgeuIscB:ωRNA complex, making it one of the smallest RNA-guided nucleases described to date.

Next, we performed further truncations to test the importance of other regions of interest. We first disrupted the A141-G59 base-stacking interaction that links P1 and P2 (Supplementary Fig. 5d) by truncating the 141-158 region (ΔP3.1). Loss of this contact abrogated DNA targeting in vivo, confirming its role in long-range structural stabilization (Fig. 3c). We then tested the effect of removing the long-range kissing loop interaction between G94-C97 and G142 and C145. The A141-G59 base stacking interaction was retained but the kissing loop was trimmed by deleting the 142-158 region (ΔP3.2). While this truncation did not prevent DNA targeting when trimmed individually, combinatorial truncations ΔP1ΔP3.2 and ΔP4ΔP3.2 significantly hindered DNA targeting, showing that this interaction is crucial for Cas9d activity (Fig. 3c). Surprisingly, disruption of base stacking interactions via purine to pyrimidine mutations in the RNA loop as well as in A141 did not hinder Cas9d activity individually (Supplementary Fig. 5e). These results highlight the importance of using high-resolution structures to inform sgRNA redesign, since long-range tertiary contacts can be critical for RNA folding.

### Cas9d has a unique mechanism for PAM recognition

We next assessed the PAM preferences of Cas9d. While Cas9d has been shown to preferentially utilize an NGG PAM for in vivo targeting, we

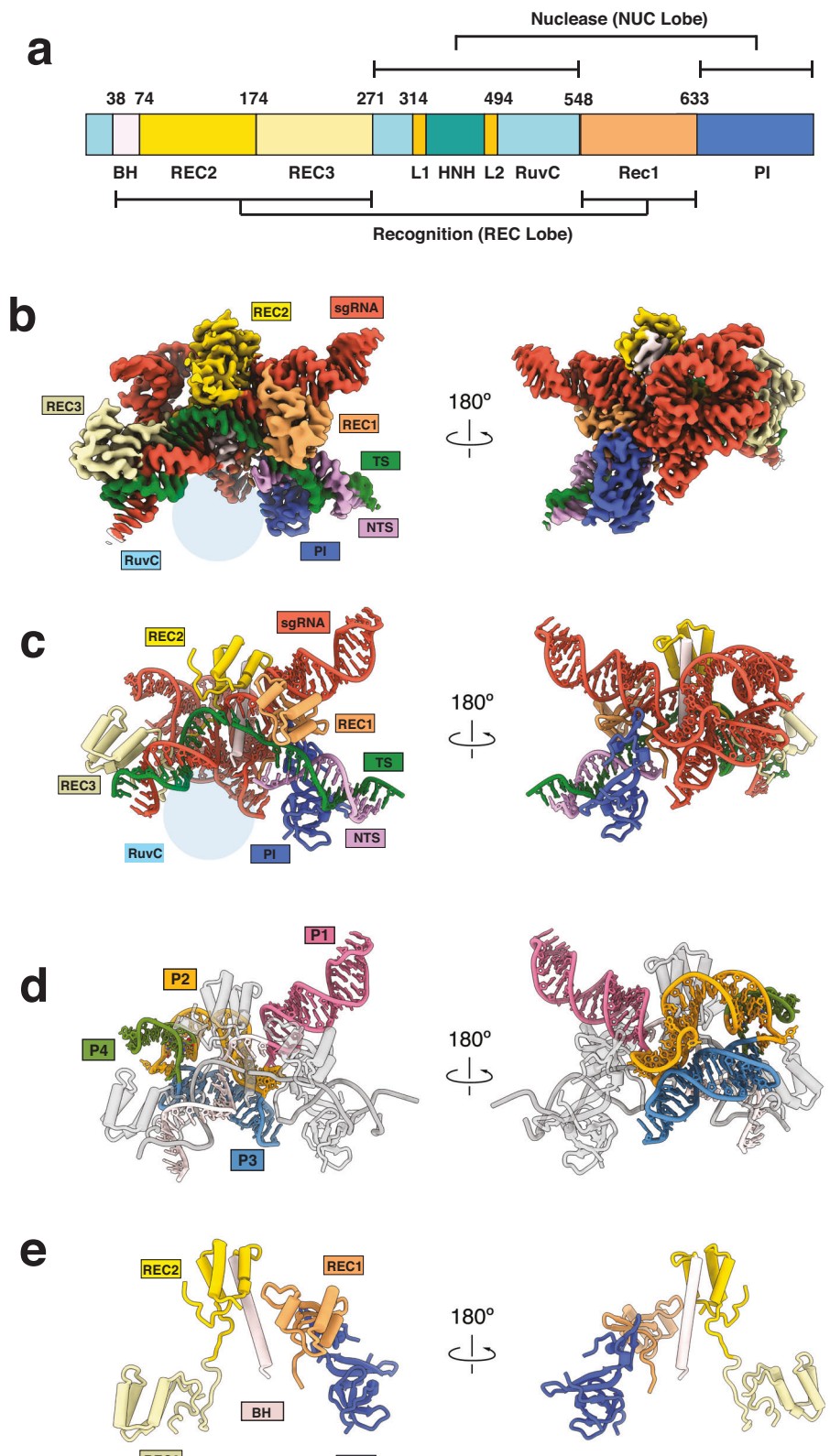

**Fig. 1 | Structures of Cas9d. a** Domain organization of Cas9d. Recognition domains REC1 (sandy brown) and REC2 (gold) are separated from REC3 (pale yellow) by a nuclease lobe comprised of RuvC and HNH domains (not resolved). PI domain (royal blue) is located on the C-terminal region of the protein. Color scheme is conserved through all Figures. **b** Front and back views of overall cryo-EM structure of Cas9d with a 20-bp R-loop. Density is colored based on proximity to modeled domains. **c** Front and back views of Cas9d atomic model with a 20-bp R-loop. **d** Front and back views of Cas9d with highlighted sgRNA. All protein and DNA are colored in gray. **e** Front and back views of Cas9d protein domains resolved in the 20bp R-loop structure.

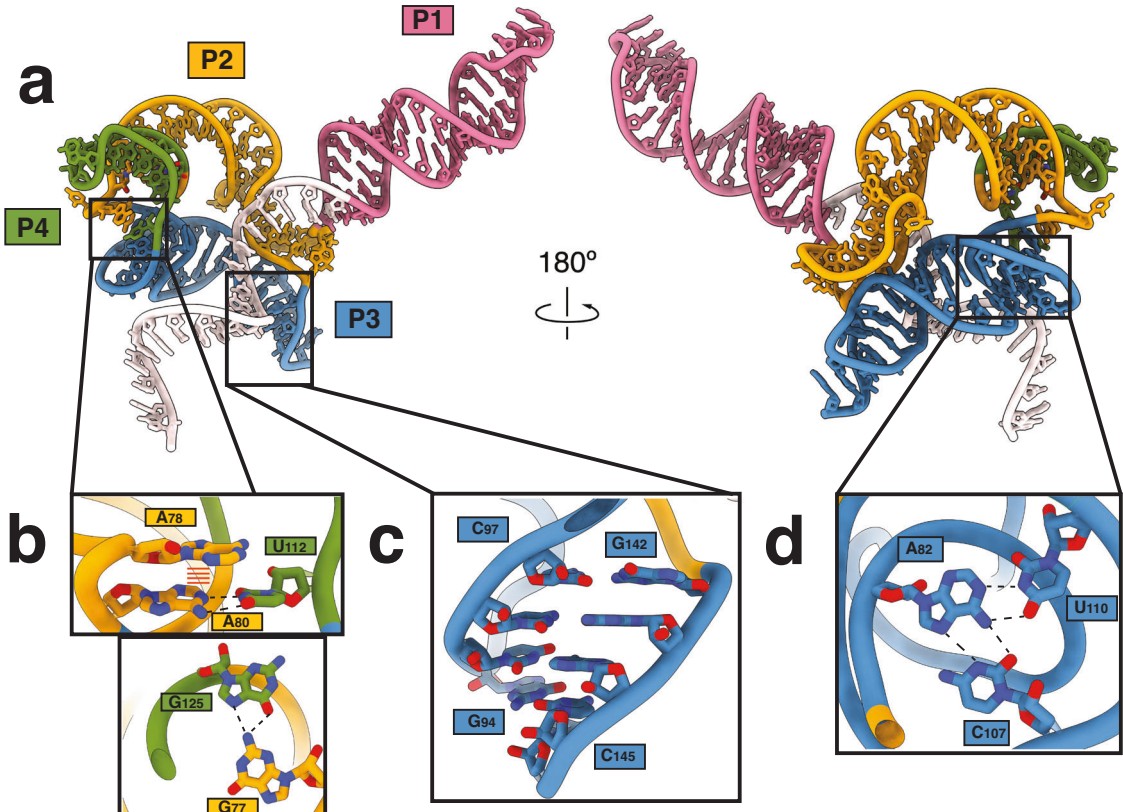

**Fig. 2 | Features of sgRNA architecture. a** Visualization of the sgRNA for Cas9d from front and back. Colored as in Fig. 1d. **b** Stabilization of sgRNA folding is driven by base stacking between A78 and the A80·U112 base pair as, well as hydrogen bonding between the G77 exocyclic amine and the C6 carbonyl group in G125. **c** Cas9d sgRNA 142-146 region forms a kissing loop interaction with the G94-C97 P3 sgRNA region. **d** Bases A82, C107, U108, form a triplex to the P3 region to fold.

found that this is not a stringent requirement in vitro, with cleavage of NGR substrates, where R is any purine. To determine the mechanism of PAM recognition, we analyzed the PAM interacting domain of our structure (Fig. 4a). It has been previously reported that most type II and type V-A nucleases employ arginine and lysine to recognize their PAMs[32,33]. Notably SpCas9 uses R1333 and R1335 to recognize the Hoogsteen face of each PAM guanosine via the formation of two H-bonds with O6 and N7 with each arginine residue through the major groove of the dsDNA duplex[34]. In contrast, Cas9d recognizes its PAM from both the major and minor grooves of the dsDNA target. From the major groove side, Cas9d employs K715 to form a contact with the N7 amine in G$_{-3}$ (Fig. 4b). More notably, on the minor groove side, Cas9d uses N651 to recognize the exocyclic amine of G$_{-2}$ and K649 to interact with the C$_{-2}$ carbonyl in the target strand (Fig. 4d). These amino acids are essential for PAM recognition, as K649A and N651A mutations completely prevent in vivo DNA targeting by Cas9d (Fig. 4f). Since N651 forms a hydrogen bond with the functional group of the C6 amine in G$_{-2}$ purines, a G$_{-2}$ to A$_{-2}$ mutation should significantly decrease PAM targeting activity. Indeed, the lack of a functional group at the C6 carbon in the adenosine base significantly reduced plasmid DNA targeting in vitro (Fig. 4g).

N654 forms a non-specific contact with the phosphate backbone of G$_{-2}$ (Fig. 4c). This is essential for DNA targeting since the N654A mutation abrogates activity (Fig. 4f). This residue contributes to DNA targeting by positioning the NTS for base-specific recognition by other residues, as previously observed for other Cas9 nucleases[35,36]. Notably, PAM recognition by Cas9d utilizes a glycine loop intertwined between the R-loop proximal region of the target and non-target strands. A G634P mutation completely prevents DNA targeting

activity, showing that this glycine loop could be important for non-target strand positioning, consistent with previous results[37,38] (Fig. 4e, f). Collectively, we find that Cas9d uses a unique mechanism to recognize an NGG PAM.

### Cas9d domain rearrangements upon DNA recognition provide multiple checkpoints before cleavage

To understand Cas9d rearrangements required for cleavage, we collected a cryo-EM dataset on the Cas9d binary (enzyme:sgRNA) complex, which yielded two distinct structures. We resolved both the Cas9d binary complex and a partial ternary complex in which 6-bps of the R-loop are formed between the sgRNA and co-purified native nucleic acid (referred as the seed conformation) both at ~3.4 Å resolution (Fig. 5a, b). We could clearly resolve the RuvC domain in the seed conformation but not in the binary or post-cleavage states (Supplementary Fig. 6a, b). The PAM-interacting domain is also unresolved in the binary complex, suggesting that it is flexible, yet stabilized upon DNA binding.

Comparison between the three states revealed that two major domain rearrangements occur during R-loop formation to facilitate the accommodation of the newly formed DNA: RNA duplex. In its binary complex, REC2 is in a downward-pointing, "closed" conformation, sterically blocking R-loop progression, as well as the placement of the scissile phosphate into the endonuclease domains. Comparison between the binary and seed states show that a REC2 region (residues 95–136) is repositioned to an upward, open conformation to allow the R-loop to propagate to completion (Fig. 5c and Supplementary Fig. 7). Deletion of this region showed no decrease in DNA targeting activity in vivo but did result in a decrease

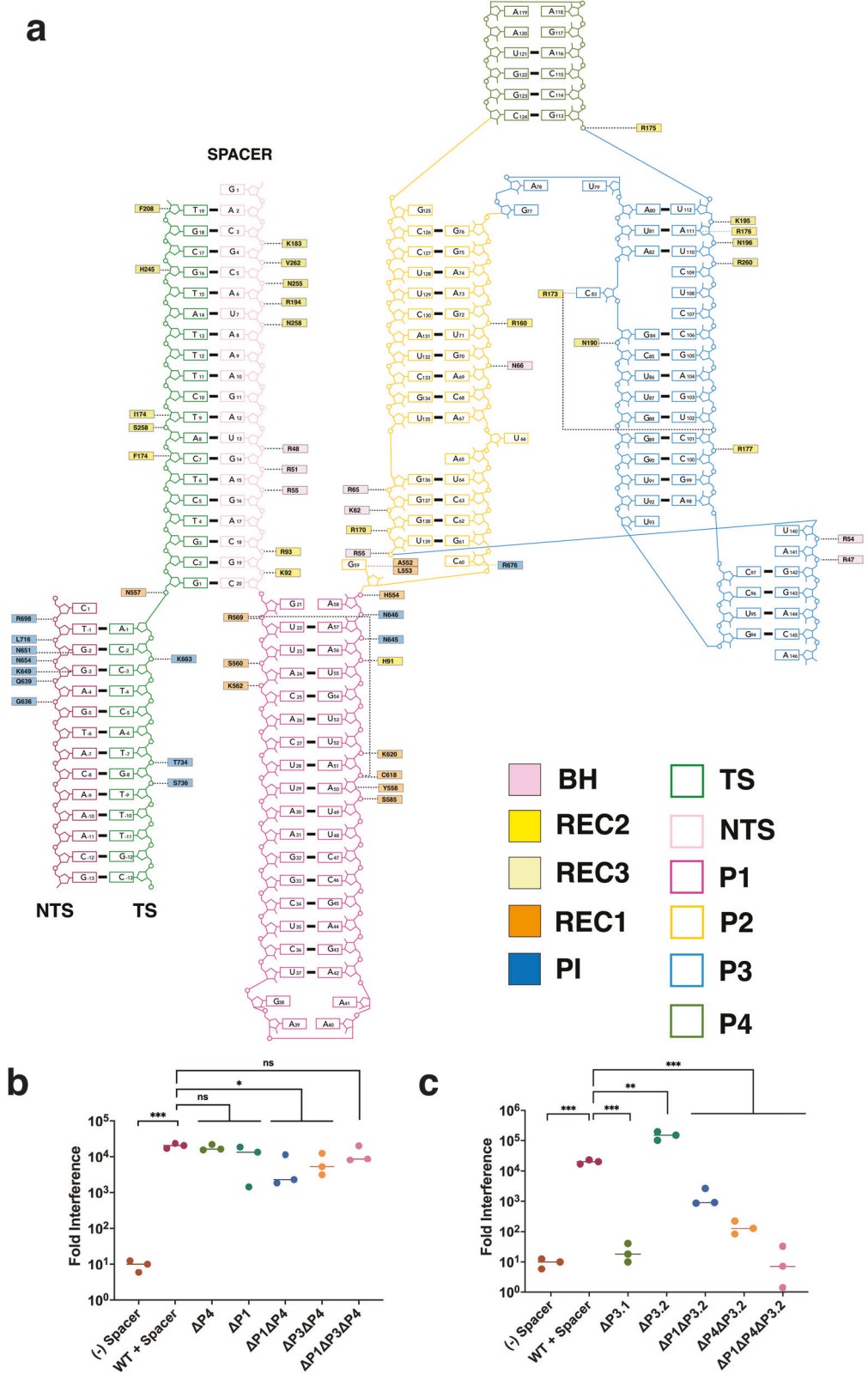

**Fig. 3 | Cas9d sgRNA engineering. a** Nucleic acid diagram for all modeled nucleotides of the Cas9d structure, including all protein contacts. The sgRNA is colored as follows: target strand (TS), forest green; non-target strand (NTS), plum; spacer, light pink; stem Loop 1, violet red; linker region, orange; stem loop 2, steel blue; stem loop 3, olive drab. **b** Nucleic acid targeting assays were developed to test activity of Cas9d with a mutated sgRNA relative to the WT sgRNA. All experiments were done in biological triplicates. Statistical significance is evaluated via two-sided unpaired $t$ tests; *$p < 0.05$, ***$p < 0.001$. Cas9d nucleic acid targeting efficiency after sgRNA trimming combinations reveals the composition of the minimal sgRNA for Cas9d. **c** Nucleic acid targeting efficiency assay reveals importance of long-range interactions for sgRNA stabilization. All experiments were done in biological triplicates. Statistical significance is evaluated via two-sided unpaired $t$ tests; **$p < 0.01$, ***$p < 0.001$. Source data are provided as a Source Data file.

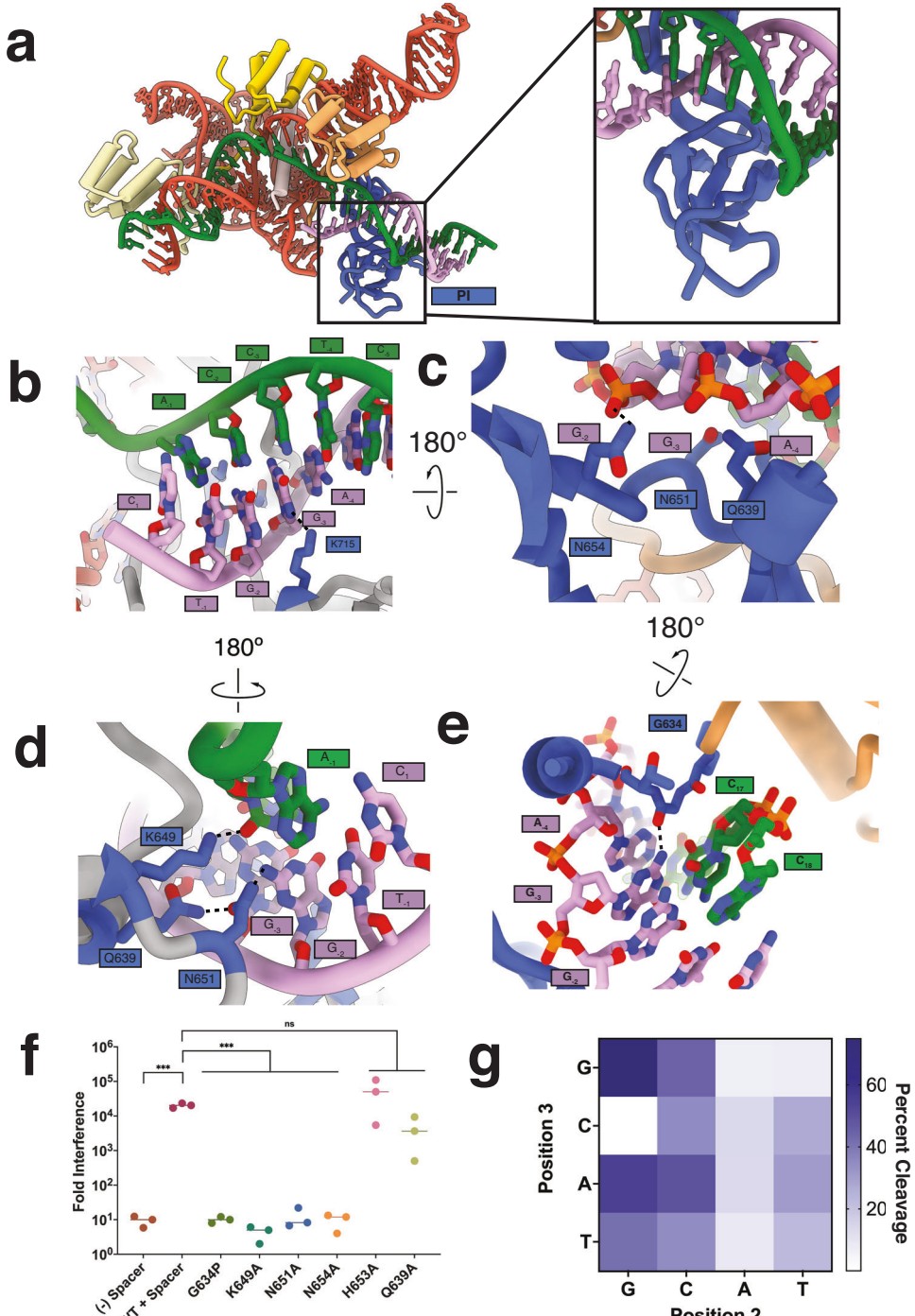

**Fig. 4 | Protospacer adjacent motif recognition by Cas9d. a** Overview of the PAM interacting domain binding to an NGG NTS PAM. **b** Recognition of an NGG PAM by Cas9d from the major groove. Hydrogen bonds are shown in dashed lines. K715 is predicted to interact with N1 in $G_{-3}$ **c** Recognition of an NGG PAM by Cas9d from the minor groove. N651 interacts with the phosphate backbone to position the NTS for PAM recognition **d** Recognition of an NGG PAM by Cas9d from the minor groove, "bottom" side. N651 and K649 interact with the C6 exocyclic amine to recognize an NGG PAM. **e** Glycine loop intercalates in between the target and non-target strands, allowing PAM recognition by Cas9d. **f** Cas9d nucleic acid targeting efficiency after mutations to the PAM interacting domain. All experiments were done in biological triplicates. Statistical significance is evaluated via two-sided unpaired *t* tests; ***$p < 0.001$. **g** Heat map plotting plasmid DNA PAM targeting activity of Cas9d in vitro. Representative of two independent experiments. Source data are provided as a Source Data file.

in the number of colonies for both induced and uninduced samples (Fig. 4f and Supplementary Fig. 8a, b), showing that deletion of this region increases the toxicity of Cas9d overexpression. One possible explanation for the increase in toxicity could be that this region may act as an inbuilt 'brake' and prevent toxic effects due to promiscuous DNA targeting. We also observed that F174 in the Cas9d seed

complex caps the R-loop between positions 6 and 7 (Supplementary Fig. 9a), akin to Y450 in SpCas9[31].

During the formation of the 20-bp R-loop from the seed complex, REC3 is displaced by ~20 Å to accommodate the bent DNA:RNA duplex and is accompanied by a ~5 Å rearrangement of the sgRNA P2 and P4 regions towards REC2, allowing REC3 to position itself in the active

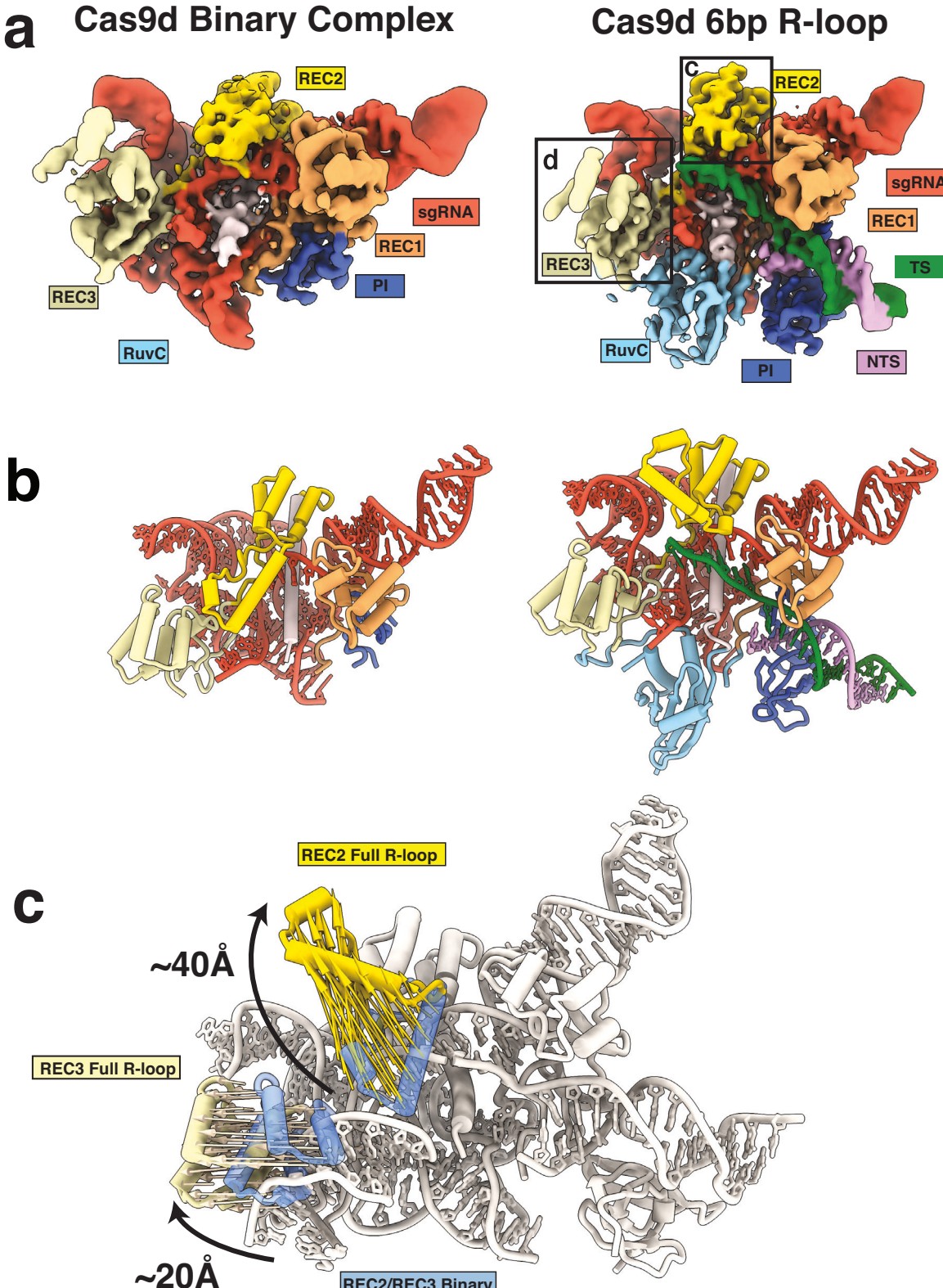

**Fig. 5 | Conformational changes guide R-loop formation by Cas9d. a** Front view of structures of Cas9d endonuclease. sgRNA binary complex (left) and 6-bp R-loop seed conformation (right). Density is colored based on proximity to modeled domains. **b** Front view of model of Cas9d:sgRNA binary complex (left) and 6-bp R-loop seed conformation (right). Density is colored based on proximity to modeled domains. **c** Comparison between binary structure and 20-bp R-loop structures show a ~40 Å REC2 96-135 region rearrangement and ~20 Å REC3 domain repositioning to interact with the R-loop.

cleavage conformation (Supplementary Fig. 9b). Similar REC2 and REC3 rearrangements have previously been reported[39,40]. The sgRNA P2-P4 rearrangement seems to be analogous to the REC2 rearrangement from the 8nt to 10nt conformations of Sp.Cas9, in which REC2 widens to allow REC3 to position itself for PAM distal R-loop recognition[31]. This along with the previously mentioned REC2 and REC3 rearrangements suggest that the seed sequence of Cas9d is bipartite, similarly to what has been reported for Sp.Cas9. These structures provide insights into how the Cas9d sgRNA can function equivalently to larger REC domains in other Cas9 orthologs.

## Cas9d enzyme kinetics reveal Cas9d undergoes an isomerization step to reach its active state

To gain mechanistic insight into target DNA binding and cleavage by Cas9d, we performed single turnover chemical quench and stopped flow kinetic assays. We first measured substrate-initiated cleavage for target and non-target strands by mixing fluorescently labeled DNA with the preformed sgRNP complex and monitoring cleavage as a function of time (Fig. 6a, b). Superficially, the observed rates appear to be much slower than rates previously reported for Sp.Cas9[41–43]. However, since the data are biphasic, they are not easily interpretable by traditional equation-based data fitting methods, because they are a complex function of R-loop formation, domain rearrangements, and endonuclease cleavage. The biphasic nature of the curves can be explained by a branched pathway where, upon DNA binding, a minor fraction (30-40%) of the sgRNP-DNA complex partitions forward to rapidly form product, while the major fraction (60-70%) goes through some other intermediate step, causing the observed rate of cleavage to be slower for the other fraction (Supplementary Fig. 10)[44].

We next measured rates of R-loop formation using a strategy employed previously[44]. We incorporated the fluorescent cytosine analog tC° at spacer position 16 in the non-target strand in the DNA and monitored unwinding (Fig. 6c). The data were again biphasic, suggesting that the partitioning of the enzyme between the active pathway and the inactive pathway occurred before R-loop formation, such that we are observing a complicated function of several steps. We then measured the rates of $Mg^{2+}$-initiated cleavage by pre-forming the sgRNA-Cas9d-DNA complex in the absence of $Mg^{2+}$ and adding $Mg^{2+}$ to initiate the reaction (Fig. 6d, e). The observed rates of cleavage were faster than seen with the substrate-initiated reaction, suggesting a slow step preceding cleavage. However, both substrate- and $Mg^{2+}$-initiated cleavage rates were considerably slower than seen for Sp.Cas9[44], suggesting that this particular Cas9 has a slower intrinsic rate of cleavage.

To provide a more rigorous interpretation of the results, we fit the data globally by simulation in KinTek Explorer using the model shown in Supplementary Fig. 10a, b[45]. The best fit to the data was obtained with a model where the DNA bound complex (ED) partitions either to form the R-loop (EDH) or to an inactive state (EDx) as with Cas9d[44]. Confidence contour analysis (Supplementary Fig. 10a) shows that the individual rates of R-loop formation are not well constrained by the data, however lower limits are obtained for both the forward and reverse rate constants, with the forward rate occurring at a rate greater than $13\,s^{-1}$. This is much faster than the rate of partitioning to the inactive state at $0.077\,s^{-1}$. To define the equilibrium constant for R-loop formation we locked the rate of R-loop formation ($k_2$) at a minimal value ($20\,s^{-1}$) and fit the rate of R-loop collapse (or reversal; $k_{-2}$) to estimate the equilibrium constant for this step. Although the forward rate of R-loop formation is fast, the equilibrium for R-loop formation is unfavorable with an equilibrium constant of 0.43. This allows the R loop step to come to equilibrium with a greater opportunity for reverse partitioning to the inactive state, which is slower to reverse at a rate of $0.01\,s^{-1}$. After forming the R-loop, the enzyme can either cleave the target strand at a rate of $0.088\,s^{-1}$ or the non-target strand at a rate

of $0.031\,s^{-1}$ in a branched pathway mechanism. The intrinsic rate constant for cleavage is approximately 100-fold slower than observed for Cas9[41].

To provide a visual aid to interpret the rate constants, we created a free energy profile for the Cas9d cleavage reaction (Fig. 6f). From the ED state, the free energy barrier for forming the R-loop (ED $\overset{\leftrightarrow}{\,}$ EDH) is lower than the barrier for partitioning to the inactive state (ED $\overset{\leftrightarrow}{\,}$ EDx), however the inactive state is more stable, as shown by the lower free energy well for this state and the slow rate of return to the ED state. The highest overall free energy barrier defines the first largely irreversible step is HNH cleavage (EDH to EDP1), indicating that specificity is determined by all reactions leading up to and including this step, giving a specificity constant defined by $K_1K_2k_3$ (Supplementary Fig. 10b). This is in contrast to Sp.Cas9 where the highest free energy barrier is for R-loop formation ($k_{-2}$) (NOTE to Printer: k2 not k-2) as the rate of R loop reversal ($k_{-2}$) is much slower than the rate of cleavage ($k_{+3}$), so the specificity constant is defined by $K_1k_2$. The slower rate of cleavage and more readily reversible R-loop formation provides a route for Cas9d to have a greater discrimination against off target DNA by allowing mismatched DNA to dissociate before being cleaved.

## Ancestral Cas9d is an active nuclease in mammalian cells

While Cas9d is promising due to its small size, members of this family have limited nuclease activity in mammalian cells[3], providing a challenge for therapeutic development. To expand the diversity of this rare family, ancestral sequence reconstruction was used to generate a putative ancestral type II-D nuclease[46,47] ancCas9d-MG-34-29 (hereby referred to as ancCas9d). AncCas9d shares an overall sequence identity with Cas9d of 79% (Fig. 7a, b), with the strongest conservation in the RuvC domain (93%) and the lowest conservation in the REC domains (71%). Notably, ancCas9d displays an expanded PAM repertoire, capable of targeting an NAG PAM-containing plasmid in vitro, while Cas9d showed minimal cleavage activity on this substrate (Fig. 7c, Supplementary Fig. 11). These differences cannot be easily explained by the Cas9d structure, as all the PAM-interacting residues identified (K649, N651, N654, and K715) are conserved. AncCas9d was also tested for nuclease activity in mammalian K562 cells via nucleofection of NGG PAM targeting guides at the AAVS1 locus and mRNA encoding the ancestral nuclease, yielding a maximum indel percentage of 56.1% (Fig. 7d).

To compare the differences in activity between ancCas9d and Cas9d, we performed in vivo fluorescence plasmid targeting assays. Successful cleavage of a GFP expression plasmid results in a decrease of GFP expression and, therefore, fluorescence detected[48]. Surprisingly, we found that ancCas9d was about 12-13 times more active than Cas9d in *E. coli* (Fig. 7e). To supplement this result, we analyzed the cleavage and R-loop formation kinetics of ancCas9d (Supplementary Fig. 12). The results indicated that although the rates of HNH and RuvC cleavage are very similar, ancCas9d favors a fully formed R-loop conformation, while Cas9d favors a pre-R-loop conformation (Table 1). Consequently, ancCas9d is more likely to adopt a catalytically active conformation than Cas9d which helps explain the increase in editing efficiency[49]. While it's not clear which residues drive the differences in activity and kinetics between ancCas9d and Cas9d, there are several differences in interface residues that may contribute. There are 15 total positions that differ between Cas9d and ancCas9d that are within 6 Å of DNA or RNA in the structure of the incomplete R loop or the full R loop (Table 2). Of these, A297T, C618Y, N644K, and R733K in ancCas9d are particularly interesting because of their properties in potentially stabilizing protein and nucleic acid interactions. Together, these results suggest that ancestrally reconstructed Cas9ds are good candidates for further structure-guided rational design to increase mammalian cell editing activity compared to modern sequences.

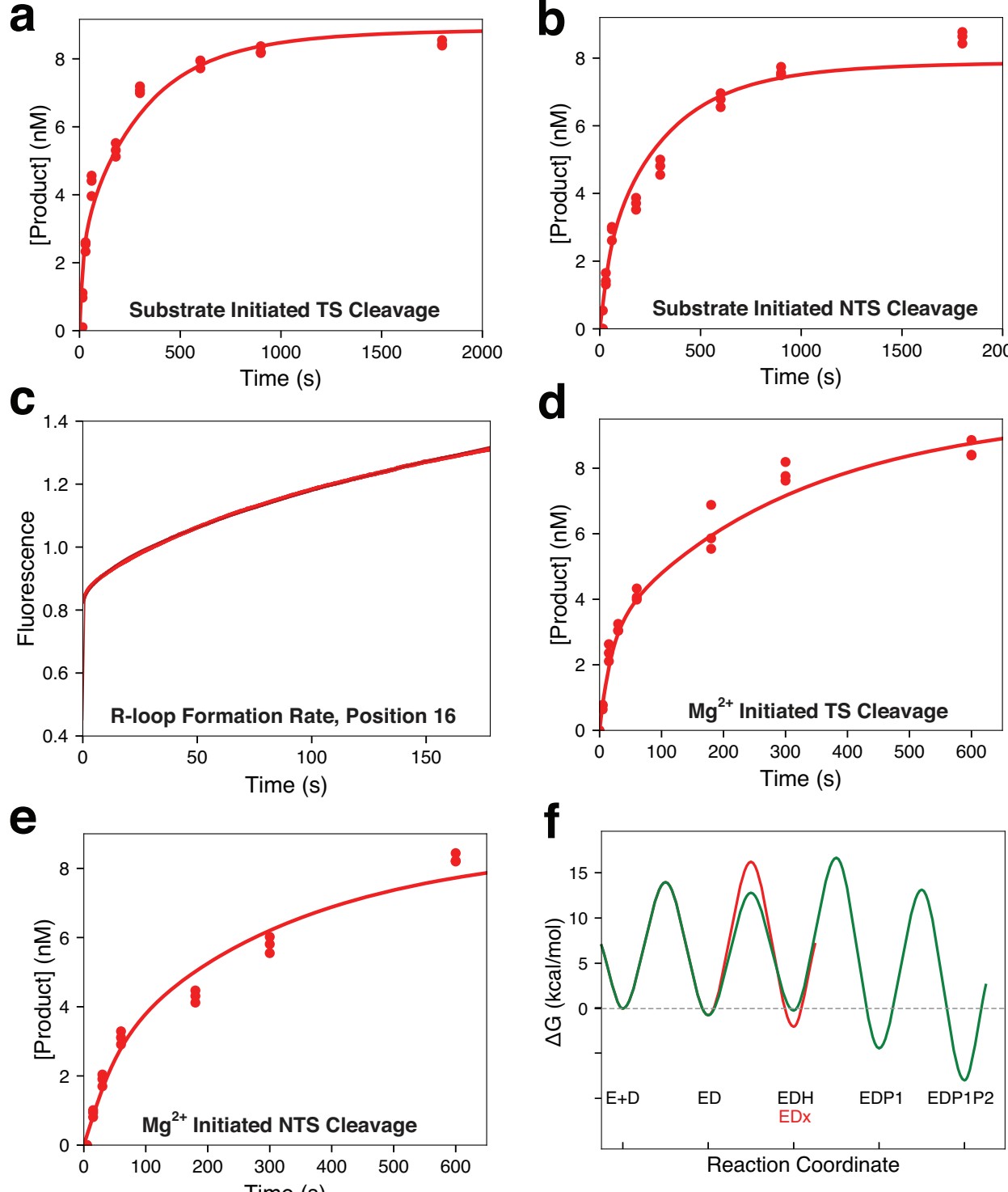

**Fig. 6 | Kinetic characterization of Cas9d reveals an inactive conformation.**
**a** Substrate initiated target strand cleavage. This experiment was initiated by mixing fluorescently labeled DNA with the sgRNP complex. **b** Substrate initiated non-target strand cleavage. This experiment was performed as in (**a**) but with a fluorescent label on the non-target strand. **c** R loop formation rates at position 16 in the spacer. The sgRNP complex was mixed with tC° labeled DNA at position 16 in the spacer and the change in fluorescence as a function of time as the R loop is formed is shown. **d** $Mg^{2+}$ initiated target strand cleavage. A ternary complex of sgRNA, Cas9d, and fluorescently labeled DNA was formed in the absence of $Mg^{2+}$,

then $Mg^{2+}$ was added to initiate the reaction and cleavage of the target strand was monitored. **e** $Mg^{2+}$ initiated non-target strand cleavage. Same experimental design as in (**d**) but with a label on the non-target strand. **f** Free energy profile for target cleavage by Cas9d. The free energy profile was generated in KinTek Explorer using the rate constants obtained (Supplementary Fig. 10). Details are given in the materials and methods section. The main pathway is shown in green, while the side path to the inactive state, EDx, is shown in red. For kinetic data in panels (**a**–**e**), the red line going through the data points is the best global fit by simulation to the model (Supplementary Fig. 10). Source data are provided as a Source Data file.

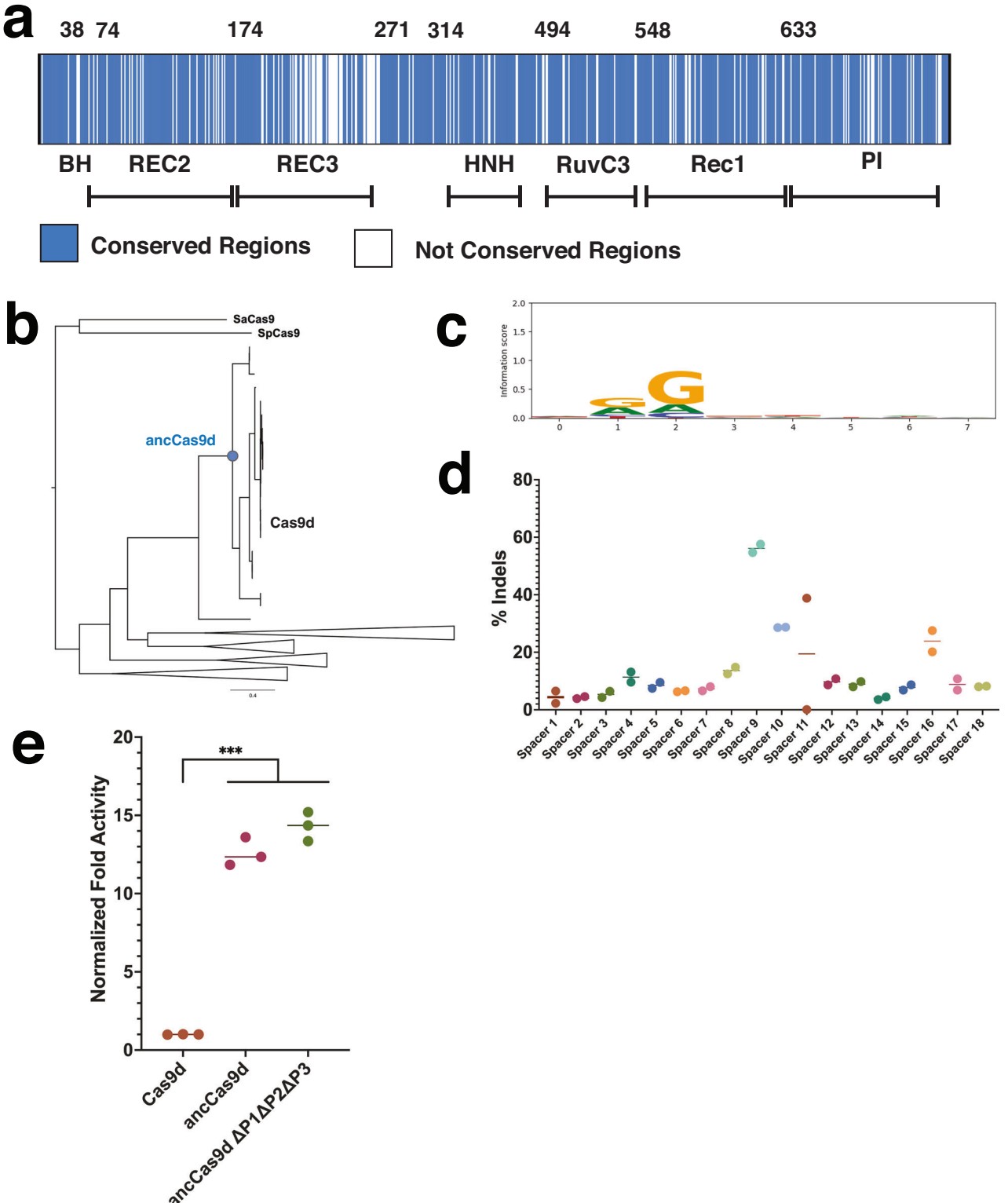

**Fig. 7 | Ancestral Cas9d reconstruction creates an enzyme capable of editing in mammalian cells. a** Cas9d to ancCas9d alignment. Blue represents conserved regions while white represents regions that are not conserved. **b** Phylogenetic tree of Cas9d protein homologs generated using MAFFT-linsi and RAxML. The circle represents the reconstructed ancestral protein. Scale bar represents substitutions per site. **c** PAM SeqLogo of ancCas9d based on NGS of cleaved plasmids from a plasmid PAM library. **d** ancCas9d editing efficiency (indel %) in K562 cells with the Cas9d sgRNA scaffold for 18 spacers targeting AAVS1 loci regions adjacent to NGG PAMs. Indel values are an average of 2 replicates. **e** GFP reporter assay; two-sided unpaired $t$ tests; ***$p < 0.001$. All experiments were done in biological triplicates. Source data are provided as a Source Data file.

**Table 1 | Table of kinetic parameters for ancCas9d and associated error estimates**

| Parameter | Best fit (lower limit, upper limit) |
|---|---|
| $k_{+x}$ | 0.0089 s$^{-1}$ (0.00887, 0.00897) |
| $k_{-x}$ | 0.0019 s$^{-1}$ (0.00188, 0.00204) |
| K$_3$ | 1.39 |
| $k_{+3}$ | 0.0131 s$^{-1}$ (0.0130, 0.0132) |
| $k_{-3}$ | 0.0094 s$^{-1}$ |
| $k_{+4}$ | 0.00875 s$^{-1}$ (0.00828, 0.00885) |
| $k_{+5}$ | 0.0099 s$^{-1}$ (0.0095, 0.0101) |

Best fit, lower, and upper limits on the kinetic parameters for ancCas9d (Supplementary Fig. 12) are shown in the table above. Lower and upper limits on the kinetic parameters are calculated at the 95% confidence interval. For comparison with Cas9d, rate constants for R-loop formation as well as the calculated equilibrium constant for this step are shown in the table.

**Table 2 | Table of amino acid mutations for ancCas9d at the protein nucleic acid interface**

| Cas9d position | Cas9d residue | ancCas9d residue | Domain |
|---|---|---|---|
| 42 | G | D | BH |
| 46 | D | G | BH |
| 244 | H | K | REC3 |
| 245 | – | D | REC3 |
| 245 | – | W | REC3 |
| 245 | I | P | REC3 |
| 248 | Q | N | REC3 |
| 258 | S | P | REC3 |
| 261 | T | L | REC3 |
| 262 | V | P | REC3 |
| 297 | A | T | RuvC |
| 618 | C | Y | REC1 |
| 632 | I | V | REC1 |
| 644 | N | K | PI |
| 733 | R | K | PI |

All residues highlighted here are residues less than 6 Å from the protein and nucleic acid interface.

## Discussion

Our work reveals the intricate architecture of a Cas9d sgRNP with DNA targeting capabilities. It also provides a basis for structure-guided sgRNA and protein engineering to develop a streamlined type II-D nuclease. We show that Cas9d contains an intricately folded sgRNA that acts as a structural scaffold to anchor flexibly tethered protein domains. Regions of this sgRNA occupy similar positions as the larger REC2 and REC3 domains of other Cas9s (Supplementary Fig. 13, 14) and also exhibit motions analogous to some of the REC domains[31], suggesting that this intricate folding is crucial for Cas9d dsDNA targeting ability.

PAM recognition is essential for self-versus-non-self-DNA recognition and has been the focus of re-engineering efforts to alter PAM preference[34,50,51]. While Cas9d has been shown to require an NGG PAM, it recognizes the guanosine bases through a distinct mechanism from other NGG-requiring Cas9 nucleases. These different strategies to achieve a common goal may also explain the promiscuous PAM targeting observed in in vitro plasmid cleavage assays. The relaxed PAM constraints of Cas9d may be a desirable feature that could be further optimized for genome editing[52].

Global fitting of the kinetic data for Cas9d required a branched step preceding R-loop formation where a fraction of the ED complex partitions into an inactive state. Based on the structures reported in this paper, we propose that this inactive state may be linked to the Rec2 domain in its downward-pointing "closed" conformation which requires a change to the open conformation to allow for R-loop propagation. Although we have modeled this as a branched pathway, it is also possible that there are two different pre-existing structural states that lead to either the active or inactive complex. Furthermore, it is intriguing that the rate of R loop formation for this enzyme is not rate limiting, in stark contrast with Sp.Cas9 and As.Cas12a[44,53]. Future kinetic studies are required to understand Rec2 repositioning and what effect this new kinetic signature has on mismatch discrimination.

This work provides the structural basis for DNA targeting by Cas9d, from PAM recognition through R-loop formation, and provides a molecular blueprint for engineering highly active, compact nucleases. With the ternary structure of Cas9d, we can further optimize this compact nuclease family through structure-based rational design, guide engineering, domain swapping, and more. These efforts could improve mammalian activity, broaden the PAM targetability, and reduce the size of the nucleases and guide. The activity of a newly constructed ancestral nuclease in mammalian cells indicates the power of ancestral reconstruction to improve activity and generate diversity within a rare family. Together, this work demonstrates the potential of this compact nuclease family for a wide range of gene editing therapeutic applications that require compact nucleases for efficient delivery.

## Methods
### Primers, RNA guides, and DNA substrates
See Supplementary Data 1.

### Cloning, expression, and purification
Cas9d protein sequence and sgRNA were obtained from the NCBI database (SRR1658473). The protein was codon optimized and cloned into a pET28 expression vector under lacO repression with a 6x N-terminal His tag and a TEV protease site (Twist Biosciences). The sgRNA was cloned into a pBR322 vector under the expression of an E. coli constitutive ribosomal promoter J23119 with a poly-T region directly downstream of the sgRNA sequence for transcription termination using KLD mutagenesis (New England Biolabs). pBR322 and pET28 expression plasmids were co-transformed into *E. coli* BL21 NICO DE3 (New England Biolabs) chemically competent cells for expression. A single colony was inoculated into 50 mL of Luria Broth (LB) under kanamycin and chloramphenicol antibiotic selection and was allowed to grow at 37 °C for 16 h. The inoculate was then back diluted 1:100-fold into two separate 1.5 L LB culture flasks and grown to OD600 0.8. Expression was induced by addition of Isopropyl-β-D-thiogalactopyranoside (IPTG) at a final concentration of 0.1 mM and growth for 16 h at 18 °C to maximize expression levels. Cells were pelleted by centrifugation (4000 × g, 60 min) and resuspended using Lysis Buffer (20 mM tris-HCl pH = 8.0, 500 mM NaCl, 10 mM MgCl2, 1 mM TCEP, 200 µM PMSF, 10% glycerol) with 1 Pierce Protease Inhibitor Tablet and 10U DNAseI. Cells were lysed by sonication and the lysate was clarified by centrifugation at 18,000 × g for 45 min at 4 °C. The supernatant was loaded into a 5 mL HisTrap HP Ni Column (Cytiva) and eluted using a gradient of Ni elution buffer (20 mM tris-HCl pH = 8.0, 500 mM NaCl, 250 mM Imidazole, 1 mM TCEP, 10% glycerol). Fractions containing the protein were combined and dialyzed with TEV protease in dialysis buffer (40 mM HEPES pH = 8.0, 150 mM NaCl, 1 mM TCEP, 10% glycerol) for 16 h at 4 °C. The sample was then purified by gel filtration chromatography (Superose 6 10/300; GE Healthcare) in dialysis buffer. Purity and quality of the protein were analyzed by SDS-polyacrylamide gels. The protein was quantified using the PIERCE Bradford assay reagent kit (Thermo Fisher) following the manufacturer's instructions.

## Cryo-EM sample preparation, data collection and processing

For the 20-bp complex, a flash frozen sample of binary (enzyme:sgRNA) Cas9d was rapidly thawed and incubated at a 10uM final concentration using a 1:1.2 enzyme:target DNA ratio at 37 °C for 1 h. For the binary and 6-bp R-loop structures no prior incubation was performed. All samples were applied to Quantifoil 1.2/1.3 400-Mesh Carbon Grids that had been glow discharged with 20 mA for 60 s. All grids were blotted using a Vitrobot Mark IV (Thermo Fisher) for 7 s, blot force 0 at 4 °C and 100% humidity, and plunge-frozen in liquid ethane. Data were collected on an FEI Titan Krios 300 V Transmission Electron Microscope (Thermo Scientific) with a Gatan BioContinuum Imaging Filter and a K3 direct electron detector. Data were collected on Serial EM v3.9 with a 0.832 Å pixel size and a defocus range of −1.5 to −2.5 μm with a total exposure time of 10 s in a total accumulated dose of 40e/A². Motion correction, contrast transfer function (CTF) estimation and particle picking was carried out in croySPARC live v4.0. A total of 4389 movies were accepted at −30° tilt for the 20-bp R-loop complex. 3202 movies were collected for the binary and 6-bp R-loop structures. All subsequent data was processed using cryoSPARC v3.2.

For the 20-bp R-loop complex 5,372,189 particles were selected using Blob Picker. A total of 3,966,068 particles were picked after a single round of 2D classification. For the 20-bp R-loop complex, a random subset of 100,000 picked particles were picked for ab-initio reconstruction (3 classes). The best class was used for heterogeneous refinement and downstream non-uniform refinement, yielding a 2.91 Å reconstruction. Micrographs were motion corrected using Motion-Cor2 followed by the same procedure outlined above, yielding a 2.73 Å structure comprised of 1,164,607 particles. For the 6-bp and binary complex structures, a total of 926,206 particles were selected after a single round of 2D classification. A 3.24 Å resolution was obtained after following the same ab-initio, heterogeneous refinement, non-uniform refinement procedure outlined above. To sort out heterogeneity, the 3.24 Å structure was subjected to a round of 3D classification and 3D variability. A 3.43 Å 6-bp R-loop complex was extracted from the 3D classification dataset. A 3.57 Å binary (Cas9d:sgRNA) complex was extracted from a 3D variability class. Final 3.37 Å and 3.40 Å structures were extracted by following the same procedure on micrographs that were motion corrected using MotionCor2.

## Model building and figure preparation

The initial backbone for the Cas9d 20-bp R-loop map was built using an automatic atomic model builder program, ModelAngelo (Cosmic2). The output from this software showed a relatively well-aligned RNA backbone as well as the alpha helices from the protein. The output was used as a fiducial for de-novo model building using a combination of coot v1.0 and ChimeraX v1.720. Once fully modeled, Isolde v1.4 was used to improve the model to map fit and for rotamer optimization. Multiple iterations of real space refinement implemented within Phenix v1.19 were carried out to improve the model geometry and map fit.

For the Cas9d binary and seed complexes, the previously build 20-bp R-loop complex was rigid body fitted into the Cryo-EM density maps. For the Cas9d seed complex, the RuvC domain was predicted via Alphafold and rigid body fitted into the density. Other newly resolved regions such as flexible side chains were build de-novo. Like in the previous model, Isolde v1.4 was used to improve the model to map fit and for rotamer optimization. Multiple iterations of real space refinement implemented within Phenix v1.19 were carried out to improve the model geometry and map fit.

## In vivo DNA targeting assays

To test for nuclease-driven DNA cleavage in vivo, *E.coli* BL21 (D3) strains (New England Biolabs) were transformed with a T7-driven plasmid and an E.coli constitutive promoter J23119 to drive sgRNA expression. The sgRNA spacer targets "CGCATAAAGATGAGACGC" in

the BL21 DE3 strain genome. Resulting colonies were inoculated in triplicate and incubated overnight in LB under antibiotic selection. Overnight cultures were then sub-cultured and grown to OD 0.6–0.8. Cultures were split into induced and uninduced cultures. Expression of the induced cultures was driven by addition of 1 mM IPTG and growth overnight at 18 °C. Nuclease efficiency was evaluated by comparing plated ten-fold spot dilutions of induced vs uninduced cultures. Growth repression was observed for isolates that successfully targeted and cleaved dsDNA. Colonies of the dilution series were quantified and compared to measure the overall reduction of growth due to nuclease-driven DNA cleavage.

## In vivo GFP reporter assays

To quantitatively assess the difference in activity between ancCas9d and Cas9d a GFP reporter assay first described by Phan et al. was utilized[48]. Cas9d endonucleases were cloned into a CloDF13 origin of replication plasmid under a pBAD promoter. To maintain origin of replication orthogonality the sgRNA was cloned separately into plasmids with a ColE1 origin of replication under pBAD expression control. A GFP expressing plasmid was cloned into a p15a low copy plasmid under a native tac promoter for continuous expression of GFP. All three plasmids were co-transformed into Dh5a cells (NEB) and plated with the appropriate antibiotics. Three separate colonies were picked the next day and were allowed to grow ON at 37 °C. 4 μL of overnight cultures were then added into 396uL LB with the appropriate antibiotics. 0.2% arabinose was added to half of the wells to compare induced vs induced samples. Samples were grown at 37 °C for 18 h while monitoring absorbance at 495 nm every 20 min using a CLARIOstar Plus plate reader (BMG Labtech). Samples were compared as a function of difference in fluorescence intensity by quantifying the total fluorescence over time.

## In vitro plasmid cleavage assays

For PAM determination, in vitro plasmid cleavage assays were used. Briefly, 16 individual plasmids were cloned containing individual PAMs at positions 2 and 3 downstream of the spacer. To evaluate PAM preference, individual plasmids containing different PAMs each normalized to a final concentration of 10 nM and were incubated with Cas9d in a 1:1 substrate:enzyme (active fraction normalized) ratio in 1x reaction buffer (150 mM NaCl, 10 mM tris pH=8.0, 10 mM MgCl2) for 30 min at 37 °C. Reactions were quenched with the addition of 0.2 μg of RNAse A (New England Biolabs) and incubation at 37 °C for 20 min, then subsequent addition of 4 units of proteinase K (New England Biolabs) and incubation at 55 °C for 30 min. Cleavage was analyzed by agarose gel electrophoresis. Band intensities were quantified using ImajeJ2 v2.14.0 and percent cleavage was determined by comparing the substrate vs product ratio in each reaction.

## Cas9 kinetic assays

6-FAM labeled TS and NTS containing dsDNA substrates were prepared by annealing with their corresponding unlabeled 55-bp ssDNA complementary oligo (Integrated DNA Technologies) by heat denaturation in 100 mM NaCl at 95 °C for 5 min followed by a quick cooldown to 4 °C. To evaluate substrate-initiated TS cleavage, Cas9d was incubated at 37 °C in 1x reaction buffer (150 mM NaCl, 10 mM Tris-HCl pH = 8.0, 10 mM MgCl$_2$) for 15 min at 37 °C. Reactions were initiated by the addition of DNA with labeled TS in a 1:1 DNA:enzyme (active fraction normalized) ratio and reactions were quenched by the addition of 250 mM EDTA at different time points. Reactions were further quenched by the addition of 0.2 μg of RNAse A (New England Biolabs) and incubation at 37 °C for 20 min, then subsequent addition of 4 units of proteinase K (New England Biolabs) and incubation at 55 °C for 30 min. Reactions were analyzed by capillary electrophoresis monitoring fluorescence emission at 520 nm. To evaluate cleavage efficiency, peak ratios of cleaved to uncleaved substrate within the same sample were

computed. To evaluate NTS cleavage, the same experiments were performed on DNA with the NTS labeled.

To evaluate $Mg^{2+}$ initiated cleavage, reactions containing a 1:1 ratio of enzyme to sgRNA were pre-incubated at 37 °C in 1x Buffer 1 (150 mM NaCl, 10 mM Tris-HCl pH=8.0, 1 mM EDTA) for 30 min followed by incubation with the DNA substrate. Cleavage was initiated by addition of a final concentration of 11 mM $MgCl_2$. Reactions were quenched as described for the substrate-initiated DNA cleavage reactions.

Stopped-flow experiments were performed at 37 °C as previously described[42]. A Cas9d sgRNA complex was incubated in a 1:1 ratio with a 55-bp duplex substrate that contained $tC°$ at position 1 or 16 in the NTS and fluorescence was monitored using AutoSF-120 stopped-flow instrument (KinTek Corporation, Austin, TX). The fluorophore was excited at 367 nm, and emission was monitored at 445 nm using a filter with a 20 nm bandpass (Semrock).

## Global fitting of kinetic data

Global data fitting was performed by fitting all experiments using KinTek Explorer v11[54] using the reaction scheme in Supplemental Fig. 10B, with experimental details of mixing steps and reactant concentrations input for each experiment. In fitting data by simulation, each experiment was modeled exactly as it was performed[45]. Chemical-quench experiments reaction products were modeled as the sum of species containing product that was labeled in the experiment (i.e. for RuvC: EDP1 + EDP1P2). The fluorescence transient in the experiment in Fig. 6c was modeled by using fluorescence scaling factors, as shown below, where $a$ scales the overall signal relative to enzyme concentration and $b$ represents the fractional change in fluorescence in forming the R-loop completed state:

$$a * (D + ED + EDx + (b * (EDH + EDP2 + EDP1 + EDP1P2))) \quad (1)$$

For the second-order DNA binding step ($k_1$), the rate was not defined by the data so the binding rate constant was locked at a conservative estimate of the diffusion limit, $100 \, \mu M^{-1} \, s^{-1}$, and the reverse rate constant was locked at $0.3 \, s^{-1}$ to give a $K_d$ for DNA binding of 3 nM, similar to prior estimates for the equilibrium constant for Cas9 binding to DNA. The binding rate constant was chosen to be much faster than subsequent R loop formation rates because there was no evidence the binding was rate-limiting.

The confidence contours shown in Supplementary Fig. 10a were derived using the FitSpace function[55] in KinTek Explorer. These confidence contour plots are calculated by systematically varying a single rate constant and holding it fixed at a particular value while refitting the data allowing all other rate constants to float. The goodness of fit was scored by the resulting $\chi^2$ value. The confidence interval is defined based on a threshold in $\chi^2$ calculated from the F-distribution based on the number of data points and number of variable parameters to give

the 95% confidence limits[45]. For the data (Fig. 6), this threshold was of 0.99 and was used to estimate the upper and lower limits for each rate constant which are reported in Table 3.

The free-energy profile in Fig. 6f was created in KinTek Explorer using the rate constants given in Supplementary Fig. 10 using simple transition-state theory.

$$rate = A * \frac{k_B T}{H} \exp\left(-\Delta G^{\ddagger}/RT\right) \quad (2)$$

where $k_B$ is the Boltzman constant, $h$ is the Planck's constant, and $R$ is the gas constant. The free-energy profile was created for the reaction at 37 °C using a transmission coefficient of $A = 0.01$ and a value for DNA concentration of 10 nM for the second order DNA binding reaction.

## Ancestral Reconstruction

Homologs of Cas9d nucleases were identified using HMM searches of Type II nucleases against a database of metagenomics-derived proteins, as previously described[3]. The resulting 190 homologs were aligned using MAFFT with parameters L-INS-i or G-INS-i and a phylogenetic tree was built using RAxML. The trees were rooted using SpCas9 and SaCas9. Sequence reconstruction was performed with the codeml package in PAML 4.8. To account for uncertainties in the phylogenies, ancestral sequences were reconstructed for both phylogenies. Insertions and deletions were identified manually for each reconstructed node.

## In vitro PAM enrichment assay

These ancestral nucleases were assayed for DNA cleavage activity via a PAM enrichment protocol, as previously described[3]. For in silico-derived ancestral nucleases which lacked contigs, sgRNA from phylogenetic relative Cas9d- was used to evaluate in-vitro activity.

Briefly, the *E. coli* codon-optimized nucleases were PCR amplified with primers binding 150 base pairs upstream and downstream from the T7 promoter and terminator sequences, respectively. For the in-vitro transcription and translation reactions in PURExpress (New England Biolabs), nuclease PCR products were added at 5 nM final concentration and expressed at 37 °C for 2 h. Following protein expression, cleavage reactions were assembled, containing a five-fold dilution of PURExpress-derived protein, 5 nM of an 8 N PAM plasmid library, and 400 nM of sgRNA targeting the PAM library, then reacted at 37 °C for 1 h.

Cleavage products were then isolated and cleaned from reactions using SPRI beads (HighPrep; Sigma-Aldrich). Recovered cleavage products were blunted via Klenow fragment and dNTPs (New England Biolabs) to prepare them for adapter ligation. Double-stranded adapter sequences were ligated to the blunted products. Adapter sites were used in conjunction with the spacer regions as primer binding sites for NGS library prep.

NGS PCR amplicons from RNA guided cleavage reactions were evaluated using D1000 automated electrophoresis (TapeStation; Agilent). Successful plasmid cleavage and PCR amplification produced bands at the ~180 bp size and were sent for NGS sequencing as previously described[3], producing a sequence logo (SeqLogo) of the filtered list of PAMs (LogoMaker).

## In vitro PAM preference assay

Ancestral nuclease activity at different PAMs were evaluated via an in-vitro cleavage assay, using similar methods as the PAM enrichment protocol above and previously described[3]. Briefly, PURExpress-derived protein was mixed in a reaction containing 1x NEBuffer 2.1 (New England Biolabs), three-fold diluted PURExpress-derived protein, 400 μM sgRNA (Cas9d), and 5 nM of target plasmids containing nAG or nGG

## Table 3 | Table of kinetic parameters for Cas9d and associated error estimates

| Parameter | Best fit (lower limit, upper limit) |
|---|---|
| $k_{+x}$ | $0.077 \, s^{-1}$ (0.065, 0.092) |
| $k_{-x}$ | $0.0099 \, s^{-1}$ (0.0084, 0.0116) |
| $K_3$ | 0.43 |
| $k_{+4}$ | $0.088 \, s^{-1}$ (0.071, 0.111) |
| $k_{+5}$ | $0.031 \, s^{-1}$ (0.027, 0.037) |

Best fit, lower, and upper limits on the kinetic parameters (Fig. 6h) are shown in the table above. Lower and upper limits for each parameter are calculated at the 95% confidence interval. Since $k_{+3}$ and $k_{-3}$ were not individually defined by the data, $k_{+3}$ was locked at a conservative value of $20 \, s^{-1}$, and the data were fit to obtain $k_{-3}$ ($47 \, s^{-1}$), which gave the equilibrium constant for this step reported in the table.

PAMs adjacent to the spacers. Cleavage reactions were incubated at 37 °C for 1 h and subsequently worked up with 0.2 mg/mL RNase A (New England Biolabs) and 4 units of Proteinase K (New England Biolabs).

Cleavage by the nucleases were evaluated using a D5000 TapeStation kit. The calibrated concentrations of the cleavage product bands (electrogram peak areas) were used to calculate percent cleavage from total plasmid band intensity using TapeStation Analysis software (Agilent).

Plasmid Cleavage % = Cleaved plasmid conc./(cleaved plasmid conc. + uncleaved plasmid conc.)*100

### Mammalian cell activity

Mammalian cell gene editing experiments were performed similarly to previously described methods[3]. In short, K562 cells were purchased from ATCC and cultured according to manufacturer procedures. Guides targeting the AAVS1 locus were previously designed based on the Cas9d guide scaffold and PAM, then synthesized by IDT. 500 ng of in-vitro transcribed ancMG34-27 or ancMG34-29 mRNA were co-transfected in 1.5E5 cells using the Lonza 4D Nucleofector (program code FF-120). Genomic extraction was performed 72 h post-transfection using QuickExtract (Lucigen). Targeted regions were amplified using Q5 High-Fidelity DNA polymerase (New England Biolabs) with target-specific primers containing NGS barcodes. PCR amplicons were analyzed with a D1000 TapeStation kit and purified with SPRI beads. Amplicon were submitted for NGS sequencing on a MiSeq (Illumina). The resulting NGS data were analyzed with an in-house indel calculator script. For guide titration experiment with ancMG34-29, various amounts (100-400 pmol) of ancCas9 guides were co-transfected with 500 ng of mRNA and processed in a similar manner as above.

### Reporting summary

Further information on research design is available in the Nature Portfolio Reporting Summary linked to this article.

## Data availability

The cryo-EM structures of Cas9d with a 20-bp R-loop, Cas9d with a 6-bp R-loop, and Cas9d:sgRNA binary complex have been deposited into the Electron Microscopy Data Bank (EMDB) with accession codes EMD-43878, EMD-43760, EMD-43757, respectively. The corresponding models have been deposited into the Protein Data Bank (PDB) with accession codes 9AUF, 8W2Z, 8W2S, respectively. Sequencing data have been deposited is the SRA with accession number PRJNA1194466. Source data are provided with this paper.

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

## Acknowledgements

We would like to thank M. Ocampo Camacho and M.F. Canedo Ocampo for assistance with the figures. We thank M. Hooper for assistance developing the GFP assay and operating the CE machine for in vitro cleavage analysis. We thank E. Schwartz and A. Brilot for expert cryo-EM support in the Sauer Structural Biology Laboratory at UT Austin. This work was funded, in part, by a sponsored research agreement with Metagenomi, Inc. (to D.W.T), a Welch Foundation Research Grant F-1938 (to D.W.T) and the Robert J. Kleberg, Jr. and Helen C. Kleberg Foundation Medical Research Grant (to D.W.T), and a grant from the National Institute of Allergy and Infectious Diseases (NIAID 1R01AI110577 to K.A.J.).

## Author contributions

R.F.O. conceptualized experiments, collected and processed Cryo-EM datasets, modeled the structures, performed biochemical, kinetic, and in vivo assays, and wrote the manuscript. J.P.K.B. collected and processed cryo-EM dataset for Cas9d 20bp R-loop complex, built initial model, conceptualized experiments, and revised the manuscript. T.L.D. performed kinetic analysis and built kinetic model for Cas9d. I.N. conceptualized and performed ancestral sequence reconstruction of ancCas9d, S.A.J. conceptualized and performed experimental work on ancCas9d. L.M.A. conceptualized experiments for ancCas9d. A.D. performed in vivo assays for Cas9d variants. S.N. performed kinetic experiments for Cas9d. K.A.J. provided guidance for the kinetic analysis for Cas9d and ancCas9d. N.C.T., C.T.B, and C.N.B provided guidance for ancCas9 experiments, D.S.A.G. designed the ancCas9d study, D.W.T. analyzed the data, edited the manuscript, provided guidance throughout, and secured funding for the work.

## Competing interests

K.A.J. is the President of KinTek, Corp., which provided the AutoSF-120 stopped-flow and the KinTek Explorer software used in this study. I.N, S.A.J, L.M.A, N.T., C.T.B, C.N.B., and D.A.S.G. are employees of Metagenomi, Inc. I.N, S.A.J, L.M.A, C.T.B, C.N.B., and D.S.A.G. are inventors on a pending patent application filed by Metagenomi, Inc. and assigned US application no. 18/669,712 based on enzymes in this work. L.M.A, C.T.B., C.N.B., and D.S.A.G are also inventors on a pending patent application filed by Metagenomi, Inc and assigned US application no. PCT/US2024/030874 based on enzymes in this work. D.W.T. has a sponsored research agreement with Metagenomi, Inc. All other authors declare no competing interests.
