## [Transparent Peer Review file · Nature Communications]

DNA targeting by compact Cas9d and its resurrected ancestor

Corresponding Author: Dr David Taylor

Version 0:

Reviewer comments:

Reviewer #1

(Remarks to the Author)

The authors present the first structure of a small Cas9d family orthologue in multiple conformational states solved by cryoEM. They observe many new unique features, such as a novel PAM recognition mode, unusual guide RNA structure, and intriguing domain flexibilities. They utilize these observations to perform structure-based truncations of the guide RNA to further minimize the complex, resulting in one of the smallest gene editor complexes known to date. The authors further perform ancestral sequence reconstruction experiments to expand the natural sequence diversity of this family.

The study is well executed, both from a structural perspective, as well as from an enzyme kinetics perspective. The engineering aspects to improve its compactness and efficiency are logical and exciting, save for one point outlined below. The text is factual and clear. I would consider trying to improve the clarity of the main figures, as their message is not intuitively clear without extensively reading the main text. I think the study will be an important addition to the field of CRISPR biology, as well as structural biology of RNA. Overall, I absolutely recommend publication after addressing the major points below and somewhat improving the figure clarity.

Major points:

1. In the last paragraph of the introduction, the authors suggest that no gene editing activity has been described for Cas9d in mammalian cells, which serves as rationale to generate related nucleases using ancestral sequence reconstruction. However, in the original Goltsman 2022 publication, the authors have demonstrated activity of a natural Cas9d orthologue in K562 cells with up to 90% editing efficiency. The current study reports a 56.1% editing efficiency of an engineered nuclease also in K562 cells at one genomic site and exhibiting suboptimal editing efficiencies at multiple other sites. While, I do find the engineering efforts interesting and they do contribute to our understanding of the diversity within this family, I think the phrasing regarding its efficiency and novelty is too strong and needs to be toned down. Additionally, gene editing activity of the WT protein presented in this work would be beneficial to be included as comparison.
2. The flexibility of a large portion of the protein is a very intriguing and unique aspect of the Cas9d clade and would be worth investigating what the consequences of such flexibility are. While there are myriad complex biophysical assays one could do, I think the easiest for the authors would be to investigate whether Cas9d is a multi-turnover enzyme (like other small Cas9 orthologues), or binds tightly like SpCas9, which may be related to this observation.
3. The section regarding the ancestral reconstruction additionally would benefit from further bioinformatic characterization of the generated sequence, for example the frequency of mutations and amino acid substitutions especially in interfaces contacting the nucleic acids. Potentially investigating whether these sequences are recovering any hallmark conserved residues of the ancestral IscB would be interesting to see, although that might be a stretch.

Minor points:

4. Going by the validation reports, the binary (Cas9d+RNA) and 6-bp complex structures have segments of the protein with very poor geometry and map inclusion scores. This is understandable with lower resolution maps, but these could still be improved and re-refined or removed if there is not sufficient density to warrant their inclusion.
5. The deposited sequences of the cryoEM structures seem to have several mutations that deviate from the reference sequence, these should be fixed unless they are intentional, in which case they should be described in the manuscript.
6. The sentence "While our Cas9d structure exhibits hallmarks of other Cas9 effectors, its unique and distinct architecture provides a structural rationale for the distinct phylogenetic clustering of type II-D effectors." is not particularly clear, I would rephrase it somewhat to highlight more the differences between Cas9d and other families.

7. Although it is stated that the structured sgRNA takes place of the REC2 domain from other Cas9 families, although it is referenced as Figure 3a. It would be nice to see such overlay for direct comparison, perhaps added to Figure 2 to fill the whitespace.

Bonus points (not required for publication, only interesting for the reviewer):

8. As the REC2 deletion might alter the checkpoint state, what are the kinetics of such a mutant?

9. Would an F174A mutant exhibit increased specificity due to seed region destabilization, as was in the case of SpCas9?

10. Would positively charged mutations in the binding channel of RuvC potentially reduce the flexibility of the domain by increasing stickiness to the substrate, and potentially increasing activity, such as was shown for small Cas12 orthologues?

Best,

Martin Pacesa

Reviewer #2

(Remarks to the Author)

This paper reports the cryo-EM structure and DNA targeting mechanism of Cas9d, a compact type II-D CRISPR-Cas effector. The authors determined three structures: a Cas9d-sgRNA-target DNA complex in its product state at ~2.7 Å resolution, a Cas9d-sgRNA binary complex, and a 6-bp R-loop structure termed the seed conformation. The latter two structures were generated from 3D classification of the Cas9d-sgRNA binary complex, and the authors interpreted that the 6-bp R-loop state is a result of co-purification of native nucleic acids. The RuvC and HNH domains are not visible in the binary and ternary states; however, the RuvC domain is observed in the seed conformation. The authors have characterized the sgRNA structure, domain organization of Cas9d, and PAM recognition. Nucleic acid targeting efficiency assays were performed to couple the structural findings. Overall, the work offers structural and mechanistic insights into the Cas9d nuclease, with cryo-EM data meeting current standards. However, there are several weaknesses. I have provided a few major and minor critiques that should be addressed.

Major points

1. The ancestral Cas9d is interesting, particularly because the authors claim that it is highly active in mammalian cells. However, the description in the Results section was very brief. Can the authors provide a possible explanation for why this construct results in higher activity in mammalian cells? Is it possible to solve the structure of this ancestral Cas9d, which may provide insights into its improved activity?

2. For Fig. S2, the authors used over 1.7 million particles for the final reconstruction. That is a lot of particles for the final reconstruction compared to cryo-EM structures of similar complexes reported in the literature. Have the authors tried performing further 3D classifications to separate possible heterogeneous conformations? For example, could they possibly identify classes where the HNH and RuvC domains can be seen? In fact, they discussed the possibility of "more generally sampling multiple conformational states" to explain the absence of nuclease domains. I highly recommend further analysis by 3D classification.

3. The authors observed a partial ternary state in the Cas9d-sgRNA complex. They referred to this as the seed conformation and interpreted that the DNA is from co-purified native nucleic acids. This is uncommon for previous cryo-EM studies on numerous Class 2 CRISPR-Cas effectors. What could be the reason for Cas9d to yield such a conformation? Furthermore, can the authors locate where this sequence is from in the genome? To be conclusive on the seed conformation, the authors should design a target DNA to mimic this state and solve the structure by cryo-EM.

Minor points

1. Labels of the figures should be improved. Some labels are omitted, while others are too small to read. The selection of colors for labels should be more carefully considered. For example, a blue background should not be combined with black labels, as it is hard to read. The figures should be labeled more clearly, even if they use the same color code, which is appreciated.

2. No citations are included in the Discussion sections.

3. In Fig. S1, I suggest the authors run a gel to show the presence of sgRNA and evaluate its length by comparing it with RNA markers.

4. The order of figure citations should be carefully edited. For example, in the first section of the Results section, after Fig. S2, the citation immediately jumps to Fig. S8.

5. In Fig. 4e, how does the glycine loop intercalate into the DNA? I cannot see it from the figure. Please clarify.

Reviewer #3

(Remarks to the Author)

Ocampo et al. report the structure of Cas9d, a type II-D CRISPR-Cas effector that is substantially smaller than other Cas9 orthologs. The structures reveal information on how Cas9d recognizes its PAM sequence and the intricate tertiary structure of the guide sgRNA. The authors also determine a model for the kinetics of DNA binding and cleavage by Cas9d. The authors perform experiments to engineer the guide RNA as well as an ancestral reconstruction that yields a version of Cas9d (ancCas9d) that can efficiently edit a genome in mammalian cells.

The structures in this work are informative and high quality. While the manuscript is generally well written, there are several errors and omissions that made it difficult to follow in places. Some conclusions are also overstated based on the evidence

provided. The authors should address the following major and minor concerns.

Major concerns:

1. The plasmid cleavage assay shown in Supplementary Fig. 1c and in the PAM cleavage assay in Fig. 4g are not described in the Methods section. How was the data in Fig. 4g generated? Was this using a plasmid library or individual plasmid sequences? What concentration(s) of Cas9d RNP were tested and what was the time point for both experiments? In Supplementary Fig. 1c, do the authors have an explanation for why the DNA is not fully linearized by Cas9d, even at the highest concentration? It also appears that the plasmid is degraded at higher Cas9d concentrations. Is this due to trans cleavage activity or a contaminating nuclease activity? Have the authors performed a control with a plasmid that lacks a target?
2. Is the difference between the deltaP3.1 and deltaP3.2 constructs a single nucleotide (A141)? It seems that both lack residues 142-158, based on the description in the results section. It is surprising that a single nucleotide can have such a large impact on the interference activity of Cas9d. Can the authors comment on whether the A141-G59 base-stacking interaction is expected to occur if A141 is at the very end of the sgRNA? If this base-stacking interaction is essential for functional sgRNA, can it be disrupted, e.g. by replacing a purine with a pyrimidine?
3. "Deletion of this region showed no decrease in DNA targeting activity in vivo but did result in a decrease in the number of colonies for both induced and uninduced samples (Fig. 4f and Supplementary Fig. 5), suggesting that it may act as an inbuilt 'brake' and prevent toxic effects due to promiscuous DNA targeting." It is unclear why Fig. 4f is referenced in this sentence. It also seems a bit speculative why deletion of this region results in fewer colonies. It is possible the protein just aggregates more and this results in toxicity. In the absence of further evidence, this statement could be softened to state that the deletion of this region increases the toxicity of Cas9d overexpression.
4. "This is reminiscent of the autoinhibitory tracr-L in *Streptococcus pyogenes*, suggesting multiple systems have independently evolved safety mechanisms to mitigate intrinsic Cas9 toxicity" The mechanism speculated by the authors is not similar to the mechanism of tracr-L, so this link seems tenuous. If the authors do include this statement, they should cite the paper that first identified tracr-L. However, given that the evidence that residues 95-136 reduce toxicity of Cas9d is very limited, it would be better to remove this sentence.
5. ancCas9d does show genome editing of >50% for one guide tested, but most of the other 17 guides had <10%. It is unclear that Cas9d has significant potential for genome editing applications. Some of the statements regarding this potential should be softened (e.g. the last sentence of the Introduction and Discussion).
6. It is unclear that ancCas9d has improved genome editing efficiency in comparison to the original Cas9d sequence. The two Cas9d's were not directly compared for the same guide RNAs. Does ancCas9d provide higher genome editing efficiency for the spacer 9 guide in comparison to the modern Cas9d? Did the authors test ancCas9d activity using guides targeting NAG PAMs?

Minor concerns

1. There are several issues with the organization of the Supplementary Figures. They are referenced in a random order in the text and multiple figures are either not cited or cited incorrectly. For example, the authors refer to similarities and differences between Cas9d and other Cas9's several times in the early sections of the results. These instances should cite Supplementary Figure 10, which is not currently cited. This figure should likely be an earlier figure (e.g. Supplementary Figure 4), as it will be cited early in the manuscript. Supplementary Figure 3 is also not cited, while Supplementary Figure 8 appears to be cited incorrectly twice in the Results section.
2. The authors used paired t-tests to compare fold interference in different strains. It would be more appropriate to use an unpaired t-test, as paired t-tests are generally used to compare measurements on the same sample (e.g. at different time points).
3. The authors have not included the sequence of ancCas9d. Please include this as supplementary information.
4. There are no references cited in the Discussion section, although the authors refer to previous work in several sentences. Please update this.
5. In the first sentence of the introduction, the authors describe a "chimeric CRISPR RNA-trans-activating crRNA", which is a good description for the sgRNA, but they never define the term sgRNA. It would be helpful to define this in this context for the sake of a broad audience.
6. In the Methods: "The inoculate was then back diluted 1:100-fold into two separate 1.5mL LB culture flasks and grown to OD600 0.8." Should this be "two separate 1.5 L Lb culture flasks"?
7. Are the panels in Figure 2 labeled correctly? It does not seem that the figure legend describes the images shown in each panel. The nucleic acid diagram is also quite small and difficult to read. It seems like this diagram could be its own figure

(potentially a Supplementary Figure), and the other panels in Figure 2 could be combined with Figure 3. This would also help the reader identify the segments of the RNA, which are labeled in Figure 2 but not in Figure 3.

8. C109 is mentioned in the legend of Fig. 3d, but it is not shown in the figure. The text also only mentions three bases and describes a triplex. Please clarify whether this structure is a base triplex or quadruplex.

9. It would be helpful if the constructs used in Fig. 3e and f could be illustrated in the figure. As it stands, the reader has to go back and forth between the description of each construct in the text and the figure.

10. The authors include two point mutations in Fig. 3f but do not comment on them in the text.

11. "These amino acids are essential for PAM recognition, as K649A and N651A mutations completely prevent in vivo DNA targeting by Cas9d." Please cite Fig. 4f at the end of this sentence.

12. "This is essential for DNA targeting since the N654A mutation abrogates activity (Fig 4e)." This sentence should cite Fig. 4f, not 4e.

13. In Fig. 4e, it is a bit unclear how/if the glycine is contacting the PAM. Is the main chain in close enough proximity to form hydrogen bonds with the PAM? It may be better to show a stick rendition of this residue, rather than a cartoon, to allow a reader to better visualize the interaction.

14. The authors state that the G634P mutation prevents DNA targeting activity, showing that this glycine is essential for non-target strand positioning. It is possible that a glycine to proline substitution might cause protein misfolding. This statement should be softened in the absence of evidence that Cas9d is properly expressed when containing this substitution.

15. "We next measured rates of R-loop formation using the same strategy we have employed previously" Should there be a citation at the end of this sentence?

16. It is difficult to compare the data for substrate-initiated and Mg²⁺-initiated cleavage shown in Fig. 6 because the timescales for the data are different. It would be helpful if the authors could provide the data on the same timescale, potentially plotted on the same graph.

17. Kd1 is described as an estimate in the Methods, but this is not clear in the description of Supplementary Fig. 9a. Please define that Kd1 was not empirically determined in the figure legend of Supplementary Fig. 9a.

18. For Supplementary Fig. 9, the legend states that rate constants derived from the fitting of kinetic data are shown in blue, but K2 is an equilibrium constant. Please clarify this in the legend or update the figure to show k⁻².

Version 1:

Reviewer comments:

Reviewer #1

(Remarks to the Author)

I thank the authors for addressing the points I raised in my previous review. I am happy with their response and require no further clarification. Congratulations on the interesting manuscript!

Reviewer #3

(Remarks to the Author)

The authors have done an excellent job of addressing my concerns, as well as the concerns of other reviewers. I believe the paper is now suitable for publication.

REVIEWER COMMENTS

Reviewer #1 (Remarks to the Author):

The authors present the first structure of a small Cas9d family orthologue in multiple conformational states solved by cryoEM. They observe many new unique features, such as a novel PAM recognition mode, unusual guide RNA structure, and intriguing domain flexibilities. They utilize these observations to perform structure-based truncations of the guide RNA to further minimize the complex, resulting in one of the smallest gene editor complexes known to date. The authors further perform ancestral sequence reconstruction experiments to expand the natural sequence diversity of this family.

The study is well executed, both from a structural perspective, as well as from an enzyme kinetics perspective. The engineering aspects to improve its compactness and efficiency are logical and exciting, save for one point outlined below. The text is factual and clear. I would consider trying to improve the clarity of the main figures, as their message is not intuitively clear without extensively reading the main text. I think the study will be an important addition to the field of CRISPR biology, as well as structural biology of RNA. Overall, I absolutely recommend publication after addressing the major points below and somewhat improving the figure clarity.

We thank the reviewer for recognizing the novelty of the structures presented. We also thank the reviewer for the helpful suggestions and additional experiments delineated in the comments. We believe that the manuscript is strengthened by the additional experiments we performed. These include the additional bioinformatic characterization of ancCas9d, as well as turnover assays to obtain more insights into the flexibility of the protein.

Major points:

1. In the last paragraph of the introduction, the authors suggest that no gene editing activity has been described for Cas9d in mammalian cells, which serves as rationale to generate related nucleases using ancestral sequence reconstruction. However, in the original Goltsman 2022 publication, the authors have demonstrated activity of a natural Cas9d orthologue in K562 cells with up to 90% editing efficiency. The current study reports a 56.1% editing efficiency of an engineered nuclease also in K562 cells at one genomic site and exhibiting suboptimal editing efficiencies at multiple other sites. While, I do find the engineering efforts interesting and they do contribute to our understanding of the diversity within this family, I think the phrasing regarding its efficiency and novelty is too strong and needs to be toned down. Additionally, gene editing activity of the WT protein presented in this work would be beneficial to be included as comparison.

We agree with the reviewer that it is important to highlight that other Cas9d families have exhibited activity in mammalian cells. We have made changes to the manuscript to reflect that this is the first member of the MG34 family that has exhibited activity in

mammalian cells. This family is particularly interesting because it is significantly more compact than other Cas9d enzymes. The changes are below:

“While the Cas9d-MG102 family of nucleases has shown potential for human genome editing applications due to their activity in mammalian cells, a significant challenge for therapeutic development has been a lack of activity in mammalian cells for the smallest Cas9d effectors. The Cas9d MG34 family is quite rare compared to other Type II nucleases and no activity has been observed in mammalian cells for the few members of this family to date. Here, we use ancestral sequence reconstruction to generate a synthetic nuclease sequence representing a putative ancestral Cas9d nuclease that exhibits nuclease activity in mammalian cells, with indel formation greater than 50% and about 200 amino acids smaller than members of the MG102 family. These results demonstrate the potential of programmable nucleases from this family to expand in vivo human genome editing applications, including by optimizing delivery through cargo-limited technologies.”

2. The flexibility of a large portion of the protein is a very intriguing and unique aspect of the Cas9d clade and would be worth investigating what the consequences of such flexibility are. While there are myriad complex biophysical assays one could do, I think the easiest for the authors would be to investigate whether Cas9d is a multi-turnover enzyme (like other small Cas9 orthologues), or binds tightly like SpCas9, which may be related to this observation.

We agree with the reviewer that the flexibility of Cas9d is interesting. To assess the flexibility of the enzyme, we performed turnover assays on linear DNA recommended by the reviewer. While Cas9d exhibited turnover at a rate of $3.4 \times 10^{-3} \text{ min}^{-1}$, we cannot conclusively state that this is a multi-turnover enzyme. It does exhibit a faster rate of turnover than Sp.Cas9 but lower than others such as Sa.Cas9.

3. The section regarding the ancestral reconstruction additionally would benefit from further bioinformatic characterization of the generated sequence, for example the frequency of mutations and amino acid substitutions especially in interfaces contacting the nucleic acids. Potentially investigating whether these sequences are recovering any hallmark conserved residues of the ancestral IscB would be interesting to see, although that might be a stretch.

We agree that the manuscript would benefit from additional bioinformatic characterization. To try to identify the residues that might enhance 34-29 activity, we identified all residues that differ between MG34-1 and MG34-29 that are within 6 angstroms of nucleotide (DNA or RNA) in either the incomplete R loop structure or the full R loop structure. We also added this to the manuscript:

“While it’s not clear which residues drive the differences in activity and kinetics between ancCas9d and Cas9d, there are several differences in interface residues that may

contribute. There are 15 total amino acids that differ between these proteins and are within 6Å of DNA or RNA in the structure of the incomplete R loop or the full R loop (Table 3). Of these, A297T, C618Y, N644K, and R733RK are particularly interesting because of their different stabilizing capabilities the protein and associated nucleic acid. These results suggest that ancestrally reconstructed Cas9ds are good candidates for further structure-guided rational design to increase mammalian cell editing activity compared to modern sequences.”

Minor points:

4. Going by the validation reports, the binary (Cas9d+RNA) and 6-bp complex structures have segments of the protein with very poor geometry and map inclusion scores. This is understandable with lower resolution maps, but these could still be improved and re-refined or removed if there is not sufficient density to warrant their inclusion.

We agree with the reviewer. We worked to further refine the model. New PDB reports are included that we hope will satisfy the reviewer.

5. The deposited sequences of the cryoEM structures seem to have several mutations that deviate from the reference sequence, these should be fixed unless they are intentional, in which case they should be described in the manuscript.

We thank the reviewer for this helpful comment. The deposited sequences for the structure are correct. Indeed, these are the sequences published for MG34-1 during the original Cas9d publication (Goltsman, Nat Comm, 2022). The sequence for Mg34-1 (Cas9d) is correct. We did not use the reference sequence, and it's unclear where that sequence came from.

6. The sentence “While our Cas9d structure exhibits hallmarks of other Cas9 effectors, its unique and distinct architecture provides a structural rationale for the distinct phylogenetic clustering of type II-D effectors.” is not particularly clear, I would rephrase it somewhat to highlight more the differences between Cas9d and other families.

We agree with the reviewer, this sentence was modified in the manuscript:

“While our Cas9d structure exhibits hallmarks of other Cas9 effectors, its unique sgRNA fold and small flexibly tethered domains provide a structural rationale for the distinct phylogenetic clustering of type II-D effectors.”

7. Although it is stated that the structured sgRNA takes place of the REC2 domain from other Cas9 families, although it is referenced as Fig. 3a. It would be nice to see such overlay for direct comparison, perhaps added to Fig. 2 to fill the whitespace.

This is a great suggestion. We have generated a new Supplementary Fig 13, which shows the overlay of REC2 and REC3 of Sp.Cas9 with the sgRNA. The R-loop for both structures is overlaid for reference. The manuscript was also modified to reflect that it occupies the space of both REC2 and REC3

Bonus points (not required for publication, only interesting for the reviewer):

8. As the REC2 deletion might alter the checkpoint state, what are the kinetics of such a mutant?

We agree with the reviewer that this is very interesting, and it will be explored in future work

9. Would an F174A mutant exhibit increased specificity due to seed region destabilization, as was in the case of SpCas9?

We agree with the reviewer that this is very interesting, and it will be explored in future work

10. Would positively charged mutations in the binding channel of RuvC potentially reduce the flexibility of the domain by increasing stickiness to the substrate, and potentially increasing activity, such as was shown for small Cas12 orthologues?

These experiments are indeed next steps.

Reviewer #2 (Remarks to the Author):

This paper reports the cryo-EM structure and DNA targeting mechanism of Cas9d, a compact type II-D CRISPR-Cas effector. The authors determined three structures: a Cas9d-sgRNA-target DNA complex in its product state at ~ 2.7 Å resolution, a Cas9d-sgRNA binary complex, and a 6-bp R-loop structure termed the seed conformation. The latter two structures were generated from 3D classification of the Cas9d-sgRNA binary complex, and the authors interpreted that the 6-bp R-loop state is a result of co-purification of native nucleic acids. The RuvC and HNH domains are not visible in the binary and ternary states; however, the RuvC domain is observed in the seed conformation. The authors have characterized the sgRNA structure, domain organization of Cas9d, and PAM recognition. Nucleic acid targeting efficiency assays were performed to couple the structural findings. Overall, the work offers structural and mechanistic insights into the Cas9d nuclease, with cryo-EM data meeting current standards. However, there are several weaknesses. I have provided a few major and minor critiques that should be addressed.

As the reviewer suggested, we have toned down some of the claims about the ancCas9d. We now claim that this enzyme is a promising nuclease for further

engineering in mammalian cells. We also agree with the reviewer that it is important to compare Cas9d and ancCas9d side by side to explain the basis for increased activity by ancCas9d. Because of this important comment, we performed side by side GFP depletion assays in *E. coli* that showed that ancCas9d is 12-13 times more active in vivo than Cas9d. Additionally, we show via kinetic characterization of ancCas9d that it favors the fully formed R-loop state, while Cas9d favors the pre-R-loop state. While we also agree with the reviewer that a structure of ancCas9d would be interesting for comparison, we believe it is outside the scope of this current work.

Major points

1. The ancestral Cas9d is interesting, particularly because the authors claim that it is highly active in mammalian cells. However, the description in the Results section was very brief. Can the authors provide a possible explanation for why this construct results in higher activity in mammalian cells? Is it possible to solve the structure of this ancestral Cas9d, which may provide insights into its improved activity?

We agree with the reviewer that this is very interesting. We have now included GFP depletion assays to show that ancCas9d is about 12-13 times more active than Cas9d, which might provide a possible explanation as to why this enzyme is more active in mammalian cells. In addition, we describe the entire kinetic pathway for ancCas9d, which shows that ancCas9d favors the fully formed R-loop complex, while Cas9d favors the pre-R-loop complex. This data is now included in Table 2.

2. For Fig. S2, the authors used over 1.7 million particles for the final reconstruction. That is a lot of particles for the final reconstruction compared to cryo-EM structures of similar complexes reported in the literature. Have the authors tried performing further 3D classifications to separate possible heterogeneous conformations? For example, could they possibly identify classes where the HNH and RuvC domains can be seen? In fact, they discussed the possibility of "more generally sampling multiple conformational states" to explain the absence of nuclease domains. I highly recommend further analysis by 3D classification.

We agree with the reviewer that additional classification could be useful to identify missing domains from the original Cas9d structure. We performed multiple rounds of 3D classification and 3D variability analysis. Unfortunately, we were unable to obtain density for either of these domains.

3. The authors observed a partial ternary state in the Cas9d-sgRNA complex. They referred to this as the seed conformation and interpreted that the DNA is from co-purified native nucleic acids. This is uncommon for previous cryo-EM studies on numerous Class 2 CRISPR-Cas effectors. What could be the reason for Cas9d to yield such a conformation? Furthermore, can the authors locate where this sequence is from in the genome? To be conclusive on the seed conformation, the authors should design a target DNA to mimic this state and solve the structure by cryo-EM.

We agree with the reviewer that this is uncommon. Most Cas9 enzymes are purified in the apo state. However, we were only able to purify Cas9d in its binary state (with sgRNA), indicating that the RNP was free to interact with any DNA while being expressed. There is a region of 6 bps of complementarity to the sgRNA in the genome that Cas9d could bind. This is a possible explanation for the partial R-loop complex. Nucleotide positions 4,216,613-4,216,618 in the *E. coli* BL21 DE3 genome (NCBI: CP053602.1) are complementary to the sgRNA and contain an NGG PAM. We believe this is an appropriate explanation for this structure and that obtaining additional structures is outside the scope of the current work.

Minor points

1. Labels of the figures should be improved. Some labels are omitted, while others are too small to read. The selection of colors for labels should be more carefully considered. For example, a blue background should not be combined with black labels, as it is hard to read. The figures should be labeled more clearly, even if they use the same color code, which is appreciated.

We agree with the reviewer that some figures could be clearer. We decreased the intensity of the PID to increase the contrast of the text relative to its background. We also added labels to Fig. 5c to increase clarity. We updated the figures to better reflect contacts between the glycine loop and the NTS. We believe that these changes have significantly improved the clarity of our figures.

2. No citations are included in the Discussion sections.

Citations have now been added to the Discussion section.

3. In Fig. S1, I suggest the authors run a gel to show the presence of sgRNA and evaluate its length by comparing it with RNA markers.

We can visualize most of the sgRNA in the structure and the enzyme is active, which we find to be more important.

4. The order of figure citations should be carefully edited. For example, in the first section of the Results section, after Fig. S2, the citation immediately jumps to Fig. S8.

We agree with the reviewer that the Supplementary Figures needed to be reordered and more clearly organized. We rearranged and organized the figures to make the manuscript clear and easy to follow.

5. In Fig. 4e, how does the glycine loop intercalate into the DNA? I cannot see it from the figure. Please clarify.

We thank the reviewer for this comment. We agree that this needed to be clarified. We redesigned Fig. 4e to show the glycine loop contacting the amine group in C2 of G₋₃ to improve visualization.

Reviewer #3 (Remarks to the Author):

Ocampo et al. report the structure of Cas9d, a type II-D CRISPR-Cas effector that is substantially smaller than other Cas9 orthologs. The structures reveal information on how Cas9d recognizes its PAM sequence and the intricate tertiary structure of the guide sgRNA. The authors also determine a model for the kinetics of DNA binding and cleavage by Cas9d. The authors perform experiments to engineer the guide RNA as well as an ancestral reconstruction that yields a version of Cas9d (ancCas9d) that can efficiently edit a genome in mammalian cells.

The structures in this work are informative and high quality. While the manuscript is generally well written, there are several errors and omissions that made it difficult to follow in places. Some conclusions are also overstated based on the evidence provided. The authors should address the following major and minor concerns.

We would like to thank the reviewer for their comments. To address this point, we have now added all the necessary methods in the manuscript, including the plasmid cleavage assay and the new in vivo GFP targeting assay that we developed to compare ancCas9d to Cas9d. It was necessary to develop this assay because we observed no colony growth when co-transforming ancCas9d with this sgRNA.

We also agree with the reviewer that the previous version of the manuscript was difficult to follow due to poor organization of the supplementary figures. We redesigned and reorganized the supplementary figures. We have also added additional labels to the main figures for clarity and improved the glycine loop interaction figure, as the reviewer suggested. We changed some of the wording in the manuscript to avoid overstating conclusions. The toxicity of the REC2 deletion mutant and the similarities to the autoinhibitory tracr-L in *Streptococcus pyogenes* has been deleted.

Major concerns:

1. The plasmid cleavage assay shown in Supplementary Fig. 1c and in the PAM cleavage assay in Fig. 4g are not described in the Methods section. How was the data in Fig. 4g generated? Was this using a plasmid library or individual plasmid sequences? What concentration(s) of Cas9d RNP were tested and what was the time point for both experiments? In Supplementary Fig. 1c, do the authors have an explanation for why the DNA is not fully linearized by Cas9d, even at the highest concentration? It also appears that the plasmid is degraded at higher Cas9d concentrations. Is this due to trans cleavage activity or a contaminating nuclease activity? Have the authors performed a control with a plasmid that lacks a target?

We thank the reviewer for pointing out this omission in the manuscript. This was a significant oversight. We included the following methods in the revised manuscript:

For PAM determination, in vitro plasmid cleavage assays were used. Briefly, 16 individual plasmids were cloned containing individual PAMs at positions 2 and 3 downstream of the spacer. To evaluate PAM preference, individual plasmids containing different PAMs each normalized to a final concentration of 10nM and were incubated with Cas9d 34-1 in a 1:1 substrate: effector (active fraction normalized) ratio in 1x effector buffer (150mM NaCl, 10mM tris pH=8.0, 10mM MgCl₂) for 30 minutes at 37°C. Reactions were quenched with the addition of 0.2ug of RNase A (New England Biolabs) and incubation at 37 °C for 20 min, then subsequent addition of 4 units of proteinase K (New England Biolabs) and incubation at 55 °C for 30 min. Cleavage was analyzed by agarose gel electrophoresis. Band intensities were quantified using ImageJ2 v2.14.0 and percent cleavage was determined by comparing the substrate vs product ratio in each reaction.

We do believe we see full linearization in Supplementary Fig. 1c, however, we do observe degradation. We believe that this was due to exonuclease contamination, since Cas9d has not been reported to exhibit cis or trans exonuclease activity.

2. Is the difference between the deltaP3.1 and deltaP3.2 constructs a single nucleotide (A141)? It seems that both lack residues 142-158, based on the description in the results section. It is surprising that a single nucleotide can have such a large impact on the interference activity of Cas9d. Can the authors comment on whether the A141-G59 base-stacking interaction is expected to occur if A141 is at the very end of the sgRNA? If this base-stacking interaction is essential for functional sgRNA, can it be disrupted, e.g. by replacing a purine with a pyrimidine?

We agree with the reviewer that it is interesting that a single nucleotide can have a dramatic effect on the activity of Cas9d. It seems like a combination of both the base pairing interactions of the P3 tetraloop and the base stacking of A141 and G59 significantly stabilize the sgRNA fold. We tested the effect of disrupting this base stacking interaction by performing a purine to pyrimidine mutation, as suggested by the reviewer, in Supplementary Fig. 5. While it did not seem to have a significant effect, it is unclear if this is because the base stacking interaction was not entirely disrupted or because only the full 141-158 deletion has a significant effect.

3. “Deletion of this region showed no decrease in DNA targeting activity in vivo but did result in a decrease in the number of colonies for both induced and uninduced samples (Fig. 4f and Supplementary Fig. 5), suggesting that it may act as an inbuilt ‘brake’ and prevent toxic effects due to promiscuous DNA targeting.” It is unclear why Fig. 4f is referenced in this sentence. It also seems a bit speculative why deletion of this region results in fewer colonies. It is possible the protein just aggregates more, and this results

in toxicity. In the absence of further evidence, this statement could be softened to state that the deletion of this region increases the toxicity of Cas9d overexpression.

The citation has been fixed and cites Supplementary Fig. 8. We also agree with the reviewer that this statement might be too speculative, and we changed the wording to:

“Deletion of this region showed no decrease in DNA targeting activity in vivo but did result in a decrease in the number of colonies for both induced and uninduced samples (Supplementary Fig. 8), suggesting that deletion of this region increases the toxicity of Cas9d overexpression. One possible explanation for the increase in toxicity could be that this region may act as an inbuilt ‘brake’ and prevent toxic effects due to promiscuous DNA targeting.”

4. “This is reminiscent of the autoinhibitory tracr-L in *Streptococcus pyogenes*, suggesting multiple systems have independently evolved safety mechanisms to mitigate intrinsic Cas9 toxicity” The mechanism speculated by the authors is not similar to the mechanism of tracr-L, so this link seems tenuous. If the authors do include this statement, they should cite the paper that first identified tracr-L. However, given that the evidence that residues 95-136 reduce toxicity of Cas9d is very limited, it would be better to remove this sentence.

We agree with the reviewer on this comment. This sentence has been removed from the manuscript

5. ancCas9d does show genome editing of >50% for one guide tested, but most of the other 17 guides had <10%. It is unclear that Cas9d has significant potential for genome editing applications. Some of the statements regarding this potential should be softened (e.g. the last sentence of the Introduction and Discussion).

We agree with the reviewer on this comment, we have modified the following statements to soften some of our conclusions:

The title for the ancestral Cas9d section has been changed to: “Ancestral Cas9d is an active nuclease in mammalian cells”

We also modified the last sentence of the ancCas9d section to: “These results suggest that ancestrally reconstructed Cas9ds are good candidates for further structure-guided rational design to increase mammalian cell editing activity compared to modern sequences.”

6. It is unclear that ancCas9d has improved genome editing efficiency in comparison to the original Cas9d sequence. The two Cas9d’s were not directly compared for the same guide RNAs. Does ancCas9d provide higher genome editing efficiency for the spacer 9 guide in comparison to the modern Cas9d? Did the authors test ancCas9d activity using

guides targeting NAG PAMs?

We agree with the reviewer that a direct comparison is necessary to state that ancCas9d is more active than Cas9d. We developed an in vivo GFP depletion assay to test the activity of ancCas9d compared to Cas9d. This is now included in Fig. 7. The results from the assay show that ancCas9d is about 12-13 times more active than Cas9d in *E. coli*, which might also provide an explanation as to why it is more active in mammalian cells.

Minor concerns

1. There are several issues with the organization of the Supplementary Figures. They are referenced in a random order in the text and multiple figures are either not cited or cited incorrectly. For example, the authors refer to similarities and differences between Cas9d and other Cas9's several times in the early sections of the results. These instances should cite Supplementary Figure 10, which is not currently cited. This figure should likely be an earlier figure (e.g. Supplementary Figure 4), as it will be cited early in the manuscript. Supplementary Figure 3 is also not cited, while Supplementary Figure 8 appears to be cited incorrectly twice in the Results section.

We agree with the reviewer that the supplementary figure order needed to be redesigned and more clearly organized. We have reorganized supplementary figures to make the manuscript clearer and easier to follow.

2. The authors used paired t-tests to compare fold interference in different strains. It would be more appropriate to use an unpaired t-test, as paired t-tests are generally used to compare measurements on the same sample (e.g. at different time points).

We would like to thank the reviewer for this comment. After further evaluation of the tests performed, we realized that we did perform unpaired t-tests. All figure labeling has been changed to reflect this.

3. The authors have not included the sequence of ancCas9d. Please include this as supplementary information.

The sequence on ancCas9d is now included as a Supplementary file.

4. There are no references cited in the Discussion section, although the authors refer to previous work in several sentences. Please update this.

Citations have been added to the Discussion section.

5. In the first sentence of the introduction, the authors describe a "chimeric CRISPR RNA-trans-activating crRNA", which is a good description for the sgRNA, but they never

define the term sgRNA. It would be helpful to define this in this context for the sake of a broad audience.

We agree with the reviewer. We have revised this to define the guide RNA:

Type II CRISPR-Cas effector nucleases associate with a single guide RNA (sgRNA), a chimeric CRISPR RNA (crRNA)-trans-activating crRNA (tracrRNA), to cleave dsDNA, targeting a region complementary to the first 18-24 nucleotides of the 5' end of the gRNA, termed the spacer.

6. In the Methods: “The inoculate was then back diluted 1:100-fold into two separate 1.5mL LB culture flasks and grown to OD600 0.8.” Should this be “two separate 1.5 L Lb culture flasks”?

We thank the reviewer for this comment. The methods section has been updated.

7. Are the panels in Figure 2 labeled correctly? It does not seem that the figure legend describes the images shown in each panel. The nucleic acid diagram is also quite small and difficult to read. It seems like this diagram could be its own figure (potentially a Supplementary Figure), and the other panels in Figure 2 could be combined with Figure 3. This would also help the reader identify the segments of the RNA, which are labeled in Figure 2 but not in Figure 3.

We thank the reviewer for this comment. Fig. 2 and Fig. 3 have been reorganized for clarity.

8. C109 is mentioned in the legend of Fig. 3d, but it is not shown in the figure. The text also only mentions three bases and describes a triplex. Please clarify whether this structure is a base triplex or quadruplex.

We have now clarified that this is a triplex that does not involve C109.

9. It would be helpful if the constructs used in Fig. 3e and f could be illustrated in the figure. As it stands, the reader has to go back and forth between the description of each construct in the text and the figure.

This was a terrific suggestion. We have now reformatted Fig. 3 to include the schematic of the gRNA, enabling readers to directly visualize both elements of the data at the same time.

10. The authors include two point mutations in Fig. 3f but do not comment on them in the text.

We thank the reviewer for this comment. We now explicitly mention it in the text:

“Surprisingly, disruption of base stacking interactions via purine to pyrimidine mutations in the RNA loop as well as in A141 did not hinder Cas9d activity”

11. “These amino acids are essential for PAM recognition, as K649A and N651A mutations completely prevent in vivo DNA targeting by Cas9d.” Please cite Fig. 4f at the end of this sentence.

We thank the reviewer for this comment. Fig. 4f has been cited after this statement.

12. “This is essential for DNA targeting since the N654A mutation abrogates activity (Fig 4e).” This sentence should cite Fig. 4f, not 4e.

We thank the reviewer for this comment. This statement now cites Fig. 4f.

13. In Fig. 4e, it is a bit unclear how/if the glycine is contacting the PAM. Is the main chain in close enough proximity to form hydrogen bonds with the PAM? It may be better to show a stick rendition of this residue, rather than a cartoon, to allow a reader to better visualize the interaction.

We agree with the reviewer. We have redesigned Fig. 4e to show the glycine loop contacting the amine group in C2 of G₋₃ to improve visualization.

14. The authors state that the G634P mutation prevents DNA targeting activity, showing that this glycine is essential for non-target strand positioning. It is possible that a glycine to proline substitution might cause protein misfolding. This statement should be softened in the absence of evidence that Cas9d is properly expressed when containing this substitution.

We agree with the reviewer that this statement could be softened. The statement was adjusted to:

“A G634P mutation completely prevents DNA targeting activity, showing that this glycine loop could be important for non-target strand positioning, consistent with previous results”

15. “We next measured rates of R-loop formation using the same strategy we have employed previously” Should there be a citation at the end of this sentence?

We thank the review for this comment. A citation has been added after this phrase.

16. It is difficult to compare the data for substrate-initiated and Mg²⁺-initiated cleavage shown in Fig. 6 because the timescales for the data are different. It would be helpful if

the authors could provide the data on the same timescale, potentially plotted on the same graph.

We thank the reviewer for this comment. While it is true that the timescales in the graphs are different, we provide rates for each step of the reaction that we believe are easier to interpret.

17. K_{d1} is described as an estimate in the Methods, but this is not clear in the description of Supplementary Fig. 9a. Please define that K_{d1} was not empirically determined in the figure legend of Supplementary Fig. 9a.

We have modified the Supplementary Figure to mention that K_{d1} was not empirically determined.

18. For Supplementary Fig. 9, the legend states that rate constants derived from the fitting of kinetic data are shown in blue, but K_2 is an equilibrium constant. Please clarify this in the legend or update the figure to show k_2 .

We have modified the legend to clarify that K_2 is an equilibrium constant.